# `UGoDIT`: Unsupervised Group Deep Image Prior Via Transferable Weights

**Shijun Liang**[* 1,2], **Ismail R. Alkhouri**[* 1,2],
**Siddhant Gautum**[2], **Qing Qu**[1], **Saiprasad Ravishankar**[2,3]
[1]Department of Electrical Engineering & Computer Science, University of Michigan - Ann Arbor
[2]Department of Computational Mathematics, Science, & Engineering, Michigan State University
[3]Department of Biomedical Engineering, Michigan State University
{shijunli,ismailal,qingqu}@umich.edu
{liangs16,alkhour3,gautamsi,ravisha3}@msu.edu

## Abstract

Recent advances in data-centric deep generative models have led to significant progress in solving inverse imaging problems. However, these models (e.g., diffusion models) typically require large amounts of fully sampled (clean) training data, which is often impractical in medical and scientific settings. Training-data-free approaches like Deep Image Prior (DIP) do not require clean images but suffer from noise overfitting and can be computationally expensive as the network parameters need to be optimized for each measurement vector independently. Moreover, DIP-based methods often overlook the potential of learning a prior using a small number of sub-sampled measurements (or degraded images) available during training. In this paper, we propose `UGoDIT` —an **U**nsupervised **G**roup **DI**P via **T**ransferable weights—designed for the low-data regime where only a very small number, $M$, of sub-sampled measurement vectors are available during training. Our method learns a set of transferable weights by optimizing a shared encoder and $M$ disentangled decoders. At test time, we reconstruct the unseen degraded image using a DIP network, where part of the parameters are fixed to the learned weights, while the remaining are optimized to enforce measurement consistency. We evaluate `UGoDIT` on both medical (multi-coil MRI) and natural (super resolution and non-linear deblurring) image recovery tasks under various settings. Compared to recent standalone DIP methods, `UGoDIT` provides accelerated convergence and notable improvement in reconstruction quality. Furthermore, `UGoDIT` achieves performance competitive with SOTA DM-based and supervised approaches, despite not requiring large amounts of clean training data. Our code is available at: UGoDIT.

## 1 Introduction

Solving inverse problems is important across a wide range of practical applications, such as those in robotics [1], forensics [2], remote sensing [3], and geophysics [4]. One class of such problems is Inverse Imaging Problems (IIPs), including super-resolution [5], image denoising [6], image deblurring [7], recovering missing portions of images [8] (in-painting), and magnetic resonance imaging (MRI) reconstruction [9].

In recent years, an abundance of deep learning (DL) methods have been developed to solve IIPs [10, 11, 12, 13, 14]. Based on the availability of training data points, DL-based IIP methods can be categorized into (*i*) data-less DL approaches, such as the ones based on the Deep Image Prior (DIP)

---

[*]Equal contribution.

39th Conference on Neural Information Processing Systems (NeurIPS 2025).

[15], Implicit Neural Representation (INR) [16], and zero-shot learning with self-supervision [17]; and (*ii*) data-centric DL approaches, such as the ones based on supervised networks [18, 19], unsupervised [20], or generative models such as those based on diffusion models (DMs) [13, 14].

Data-centric methods rely on pre-trained models that typically require large training sets. In the context of scientific or medical IIPs, in addition to being large-scale, this translates to the need for fully sampled measurements—i.e., clean images (serveing as ground truth). Unlike natural images, acquiring large datasets of scientific or medical data such as fully sampled MRI scans is particularly challenging [21, 22, 23]. The underlying data may be inherently undersampled or corrupted with unknown noise. Similar issues arise in other domains such as medical computed tomography (CT) [24]. To mitigate this, previous DM-based methods for MRI and CT often fine-tune DMs pre-trained on natural images using a limited number of fully sampled medical images [13, 12, 14]. This underscores the need for approaches that reduce reliance on large, clean (or fully sampled) datasets.

Training-data-free methods, such as DIP [15], which rely on untrained networks (e.g., U-Net [25]), avoid the need for training data altogether—fully sampled or otherwise. However, they often face two main challenges: (*i*) vulnerability to noise overfitting (though recent variants, such as auto-encoding sequential DIP (aSeqDIP) [26], have shown improved robustness), and (*ii*) high computational cost at inference time (as a full set of network parameters need to be optimized for each measurement independently), especially compared to the inference time of supervised methods. Although recent work, such as MetaDIP [27], has explored transferring learned DIP weights to speed up inference, it still requires supervised training with fully sampled data points, and does not achieve competitive results when compared to standalone DIP methods as was demonstrated in [28].

The scarcity of fully-sampled measurements (i.e., clean images as ground truth) in certain applications, and the inability of most DIP methods to transfer learned weights serve as our motivation to explore the following question: *Is it possible to train a fully unsupervised model using only a few degraded images, such that the learned weights can later be used to reconstruct unseen measurements during inference?* We term the regime when only very few training images are available as the "low-data regime", and towards answering the above question, we make the following contributions:

- **Decoder-disentangled architecture for training**: Given a few measurement vectors (or degraded images), we propose the use of a shared encoder and multiple decoders for each measurement vector during training such that the inputs, shared encoder, and multiple decoders are optimized jointly using a measurement-consistent and auto-encoded objective function.

- **Transferable weights at testing**: Given an unseen measurement vector, we transfer the pre-trained weights of the shared encoder to a single-encoder, single-decoder DIP network. Reconstruction is then performed by fixing the encoder and optimizing the decoder parameters using an input-adaptive, measurement-consistent objective function. Therefore, we term our method as **U**nsupervised **G**r**o**up **DI**P with **T**ransferable weights (`UGoDIT`).

- **Extensive evaluation**: We conduct extensive evaluations using multi-coil MRI, super resolution, and non-linear deblurring. We show that we outperform most recent advanced DIP methods while performing comparably with DM-based and supervised data-intensive methods all while requiring only a very few degraded images. Furthermore, our ablation studies show the impact of each component such as the proposed disentangled decoders.

## 2  Preliminaries & related work

Inverse Imaging Problems (IIPs) are defined as recovering an image $\mathbf{x}^* \in \mathbb{R}^n$ from measurement vector (or degraded image) $\mathbf{y} \in \mathbb{R}^m$ (with $m \leq n$), governed by the forward operator $\mathbf{A}$. For multi-coil MRI, the forward operator is given as $\mathbf{A} = \mathbf{MFS}$, for which $\mathbf{M}$ denotes coil-wise undersampling, $\mathbf{F}$ is the coil-by-coil Fourier transform, and $\mathbf{S}$ represents the sensitivity encoding with multiple coils[2]. For natural image restoration tasks, the forward operator in super resolution (SR) is a down sampler. For non-linear deblurring (NDB), the non-linear forward operator is the neural network approximated kernel given in [7] and adopted in multiple works such as [14, 29].

Next, we provide a brief overview of closely related DL-based image reconstruction methods based on the availability of training data.

---

[2]In MRI, the entries of $\mathbf{x}^*$ and $\mathbf{y}$ contain both real and imaginary numbers. However, to generalize the definition of the problem, we use real numbers.

**High-data regime methods:** Training data-intensive methods—that is, methods that rely on a large number of training data points—can be categorized into generative, supervised, and unsupervised approaches. Among generative methods, recently, numerous IIP solvers have employed pre-trained diffusion models (DMs) as priors (e.g., [13, 14]; see [30] for a recent survey). These methods typically modify the reverse sampling steps to enable sampling from a measurement-conditioned distribution. In supervised end-to-end (E2E) approaches, a variety of techniques have been explored, including unrolled networks such as the variational network (VarNet) in [18]. A common limitation of both generative and supervised methods is their reliance not only on large amounts of training data, but also on clean (i.e., fully sampled) ground truth images—an assumption that may not hold in many medical and scientific applications such as MRI reconstruction[3]. Moreover, DM-based generative methods are generally slower at inference compared to supervised counterparts. Unsupervised (or self-supervised) methods have also been explored [32, 33, 34]. While they do not require clean images, they still depend on a large amount of training data and often fall short of the performance achieved by supervised or generative models [35]. In this paper, we introduce a method that requires neither large-scale training data nor clean images (or fully sampled measurements).

**Training-data-free DIP-based methods:** The pioneering work of [15] introduced the Deep Image Prior (DIP), a fully training-data-free approach for solving IIPs, demonstrating that an untrained convolutional neural network (CNN) can effectively recover low-frequency image structures using the implicit bias of the network. However, DIP is prone to noise overfitting due to the network's over-parameterization. As a result, numerous techniques have been proposed to mitigate this issue, including the use of regularization [36, 26], early stopping [37], and network re-parameterization [28, 38]. We refer the reader to the recent tutorial in [39] for a comprehensive overview of DIP and its variants. Most DIP methods require optimizing a full set of weights for each measurement vector independently. In this paper, we present a DIP-based method that learns a subset of transferable weights from only a few degraded images, addressing this gap.

**Low-data regime methods:** Low-data regime refers to the case where the number of training images is extremely small for training standard supervised, generative, and/or unsupervised methods. Here, we discuss two methods that operate in the low-data regime and are closely related to our work. The first is MetaDIP [27], where the authors leverage meta-learning to learn transferable weights for accelerated DIP reconstruction at test time. MetaDIP adopts a standard single-encoder, single-decoder U-Net architecture. There are three key differences between UGoDIT and MetaDIP: (*i*) MetaDIP requires clean images during training, while our method uses only degraded images (or sub-sampled measurement vectors); (*ii*) MetaDIP does not include architectural changes during training; (*iii*) their evaluation is limited to natural image restoration tasks, whereas our method also considers more complex, large-scale MRI reconstruction. The second method is STRAINER [40], which extends the Implicit Neural Representation (INR) framework [41, 42] to the low-data regime. Specifically, the authors propose using a fully connected network (or multi-layer perceptron, MLP) with multiple output sub-networks to learn transferable weights. At test time, a subset of these weights is used to initialize a subset of the MLP-based INR model for image recovery. The main differences between UGoDIT and STRAINER are: (*i*) STRAINER uses MLPs with spatial coordinate input embedding, while our method is based on CNNs and DIP; (*ii*) STRAINER uses clean (fully sampled) training data, whereas UGoDIT uses degraded images (or sub-sampled measurements); (*iii*) the learned weights in STRAINER are used only for *initializing* part of the test-time INR network, whereas in our method, we keep the parameters of the pre-trained shared encoder unchanged during testing and only adapt the parameters of the test-time decoder.

## 3 Proposed method

Assume that we are given a set $Y$, consisting of $M$ under-sampled measurement vectors (or degraded images), where each $\mathbf{y}_i \in Y$, for $i \in [M] := \{1, \ldots, M\}$, is an $m$-dimensional vector. The degraded images in set $Y$ are obtained using the following forward models:

$$\mathbf{y}_i = \mathbf{A}_i \mathbf{x}_i^* + \boldsymbol{\eta}_i , \tag{1}$$

---

[3]We note that there have been recent efforts to train DMs on degraded images, such as the study in [31]. However, these approaches still rely on large amounts of training data.

where $\boldsymbol{\eta}_i$ represents the noise in the measurement domain, e.g., assumed sampled from a Gaussian distribution $\mathcal{N}(\mathbf{0}, \sigma_{\mathbf{y}_i}^2 \mathbf{I})$, with $\sigma_{\mathbf{y}_i} > 0$ representing the noise level of the $i$-th vector. In this setting, we assume access only to $\mathbf{y}_i$ and the corresponding $\mathbf{A}_i$ for each $i \in [M]$. We note that while we adopt a linear forward model for notational simplicity, our approach can be applied to nonlinear IIPs, as we will demonstrate in our experimental results.

Here, the number of available training images, $M$, is extremely small (e.g., $M < 10$), making it infeasible to learn a data distribution to be used as a prior during inference. Under this under-sampled (or degraded), low-data regime, we consider the following question:

> Given a CNN architecture and a set of $M$ measurement vectors, $\mathbf{y}_i \in Y$, with corresponding forward operators $\mathbf{A}_i$, can we learn a set of weights such that, for a unseen test-time measurement vector $\mathbf{y} \notin Y$ and its operator $\mathbf{A}$, we can reconstruct the image both *robustly* and *efficiently*?

Here, *robust* reconstruction refers to achieving high-quality recovery while being resilient to noise overfitting—an issue commonly encountered in standalone DIP methods [39, 36][4]. *Efficient* reconstruction refers to faster convergence compared to standalone DIP.

Among existing DIP methods, autoencoding Sequential DIP (aSeqDIP) [26] has demonstrated superior performance in terms of robustness and reconstruction quality. Meanwhile, Early Stopping DIP (ES-DIP) [37] and Deep Random Projector (DRP) [28] have shown improved efficiency in terms of faster run-time.

To enable the learning of transferable weights–which is the central goal of this paper– we propose the use of a modified network architecture: a shared encoder jointly learned across all $M$ degraded images, along with a separate decoder for each individual measurement vector $\mathbf{y}_i$, resulting in $M$ disentangled decoders. To mitigate noise overfitting, we adopt an input-adaptive autoencoding objective, which we describe in the next subsection. At test time, given an unseen vector $\mathbf{y}$, we use a DIP network, consisting of the learned shared encoder and a randomly initialized decoder, and perform image reconstruction by optimizing only the decoder parameters. In other words, given $\mathbf{y}$ at test time, the pre-trained encoder is used to extract latent features and the decoder parameters are optimized to adapt to the unseen measurement vector.

We note that using a single encoder with multiple decoders has also been explored in other areas of machine learning, including multi-task and federated learning [43, 44]. Another example is [45], which proposed a multi-decoder architecture to accelerate the training of diffusion models.

### 3.1 Proposed training optimization and network architecture

Assume we are given a CNN network (e.g., a U-Net [25]). Let $h : \mathbb{R}^n \to \mathbb{R}^l$, parameterized by $\phi$, denote the encoder function of the CNN–i.e., the **downsampler**, consisting of a stack of convolution and pooling layers that extracts and compresses features, with $l \ll n$. Similarly, let $g : \mathbb{R}^l \to \mathbb{R}^n$, parameterized by $\psi$, be the decoder function–i.e., the **upsampler**, comprising a stack of transposed convolution and/or interpolation layers that reconstruct the high-resolution output from the compressed representation.

Given some input $\mathbf{z} \in \mathbb{R}^n$, the network output is expressed as $g_\psi \circ h_\phi(\mathbf{z})$. Using this setting with some measurement vector $\mathbf{y}$ and its forward operator $\mathbf{A}$, reconstruction via vanilla DIP [15] is formulated as in (2), where $\hat{\mathbf{x}}$ denotes the reconstructed image.

$$\hat{\phi}, \hat{\psi} = \underset{\phi, \psi}{\operatorname{argmin}} \|\mathbf{A} g_\psi \circ h_\phi(\mathbf{z}) - \mathbf{y}\|_2^2, \qquad \hat{\mathbf{x}} = g_{\hat{\psi}} \circ h_{\hat{\phi}}(\mathbf{z}) . \tag{2}$$

Given a set of $M$ degraded measurements of images, $\mathbf{y}_i \in Y$, each associated with a forward operator $\mathbf{A}_i$ for $i \in [M]$, the proposed `UGoDIT` method utilizes a shared encoder $h_\phi$ and a distinct decoder $g_{\psi_i}$ for each training image. This results in a total of $M$ disentangled decoders tailored to the individual measurement vector.

Given that we have $M$ measurement vectors, choosing the input is not straightforward across a training set. Hence, we develop an input-adaptive approach. To this end, during training, we propose

---

[4]Standalone DIP refers to the standard setting where a single measurement vector is used, and the network is optimized from randomly initialized parameters.

**Algorithm 1 Training:** **U**nsupervised **G**roup **DI**P **T**ransferable via weights (UGoDIT)

---

**Input**: Number of training images $M$, measurement vectors $\mathbf{y}_i$ and forward operators $\mathbf{A}_i$ for $i \in [M]$, number of input updates $K$, number of gradient updates $N$, regularization parameter $\lambda$, and learning rate $\beta$.

**Output**: Trained encoder weights $\hat{\phi}$.

**Initialization**: $\mathbf{z}_i \leftarrow \mathbf{A}_i^H \mathbf{y}_i$ ; $\phi, \psi_i \sim \mathcal{N}(\mathbf{0}, \sigma_{\text{ini}}\mathbf{I})$ for $i \in [M]$.

1: **For** $K$ iterations, **Do**

2:     **For** $N$ iterations, **Do** (Shared encoder and $M$ decoders parameters update)

3:         $\psi_i \leftarrow \psi_i - \beta \nabla_{\psi_i} \Big[ \|\mathbf{A}_i g_{\psi_i} \circ h_\phi(\mathbf{z}_i) - \mathbf{y}_i\|_2^2 + \lambda \|g_{\psi_i} \circ h_\phi(\mathbf{z}_i) - \mathbf{z}_i\|_2^2 \Big], \forall i \in [M]$

4:         $\phi \leftarrow \phi - \beta \nabla_\phi \Big[ \sum_{i \in [M]} \|\mathbf{A}_i g_{\psi_i} \circ h_\phi(\mathbf{z}_i) - \mathbf{y}_i\|_2^2 + \lambda \|g_{\psi_i} \circ h_\phi(\mathbf{z}_i) - \mathbf{z}_i\|_2^2 \Big].$

5:     **Obtain** $\mathbf{z}_i \leftarrow g_{\psi_i} \circ h_\phi(\mathbf{z}_i)$. (Network input update for each training image $i \in [M]$)

6: **Trained weights:** $\hat{\phi} \leftarrow \phi$

---

optimizing the weights of the shared-encoder, decoder-disentangled CNN architecture using the following objective:

$$\hat{\phi}, \hat{\psi}_i, \hat{\mathbf{z}}_i = \operatorname*{argmin}_{\phi, \psi_i, \mathbf{z}_i, i \in [M]} \Big\{ \sum_{i \in [M]} \|\mathbf{A}_i g_{\psi_i} \circ h_\phi(\mathbf{z}_i) - \mathbf{y}_i\|_2^2 \text{ s.t. } g_{\psi_i} \circ h_\phi(\mathbf{z}_i) = \mathbf{z}_i, \ \forall i \in [M] \Big\} . \quad (3)$$

To solve (3), we initialize $\phi, \psi_i \sim \mathcal{N}(\mathbf{0}, \sigma_{\text{ini}}^2 \mathbf{I})$ and set $\mathbf{z}_i \leftarrow \mathbf{A}_i^H \mathbf{y}_i$ for all $i \in [M]$, where $\sigma_{\text{ini}}$ corresponds to the standard deviation for initialization. We then perform the following alternating updates for $K$ iterations (which corresponds to the number of network inputs updates):

$$\phi, \psi_1, \ldots, \psi_M \leftarrow \operatorname*{argmin}_{\phi, \psi_i, i \in [M]} \Big[ \sum_{i \in [M]} \|\mathbf{A}_i g_{\psi_i} \circ h_\phi(\mathbf{z}_i) - \mathbf{y}_i\|_2^2 + \lambda \|g_{\psi_i} \circ h_\phi(\mathbf{z}_i) - \mathbf{z}_i\|_2^2 \Big] , \quad (4a)$$

$$\mathbf{z}_i \leftarrow g_{\psi_i} \circ h_\phi(\mathbf{z}_i), \forall i \in [M] , \quad (4b)$$

where $\lambda$ is a regularization parameter. Similar to aSeqDIP [26], we alternate between two phases: (*i*) perform $N \ll K$ gradient updates to optimize the network parameters $\phi$ and $\psi_i$ using the optimization in (4), and (*ii*) update the network inputs $\mathbf{z}_i$ via forward passes. Specifically, each input $\mathbf{z}_i$ is updated by passing it through the shared encoder $h_\phi$ and the corresponding decoder $g_{\psi_i}$.

In the training optimization of (4), the first term enforces the output of decoder $i$ to be consistent with the measurements vector $\mathbf{y}_i$ governed by $\mathbf{A}_i$. The second term is an autoencoding term that implicitly enforces the equality constraints in (3). It is used to mitigate noise overfitting by enforcing that the output of each decoder is approximately consistent with the updated input of that particular image $\mathbf{z}_i$ as the optimization progresses.

In what follows, we provide a justification of how we relaxed (3) into the alternating optimization updates in (4). The general intuition stems from prior studies that highlight the impact of network inputs in DIP-based reconstruction [46, 26, 36], where the study in [26] shows that reconstruction quality improves when the network input is closer to the ground truth.

Assume that the iterates in (4) converge to $\phi^*, \psi_i^*, \mathbf{z}_i^*, \forall i \in [M]$. Then, according to the second equation in (4), for continuous mappings $g_{\psi_i} \circ h_\phi(\cdot), \forall i \in [M]$, we have $\mathbf{z}_i^* = g_{\psi_i^*} \circ h_{\phi^*}(\mathbf{z}_i^*)$. Substituting this into the first equation of (4), in the limit, we get

$$\phi^*, \psi_i^* = \operatorname*{argmin}_{\phi, \psi_i, i \in [M]} \Big\{ \sum_{i \in [M]} \|\mathbf{A}_i g_{\psi_i} \circ h_\phi(\mathbf{z}_i^*) - \mathbf{y}_i\|_2^2 \quad \text{s.t.} \quad g_{\psi_i} \circ h_\phi(\mathbf{z}_i^*) = \mathbf{z}_i^*, \quad \forall i \in [M] \Big\} ,$$

which corresponds to the minimizer of (3) given a fixed $\mathbf{z}_i^*$. For each decoder, the limit points of (4) correspond to the solution of a constrained version of vanilla DIP in (2). The constraint enforces additional prior that could alleviate overfitting. While it is not straightforward to use first-order gradient-based algorithms for solving (3) given the hard equality constraints, the limit points in the updates of (4) nevertheless minimize (3). Furthermore, the network input is updated by a sequential feed forward process without needing expensive gradient-based updates.

The training procedure of UGoDIT is given in Algorithm 1. In the initialization step, we assume the encoder and decoder weights are of equal size, though this is not necessary a requirement. We note

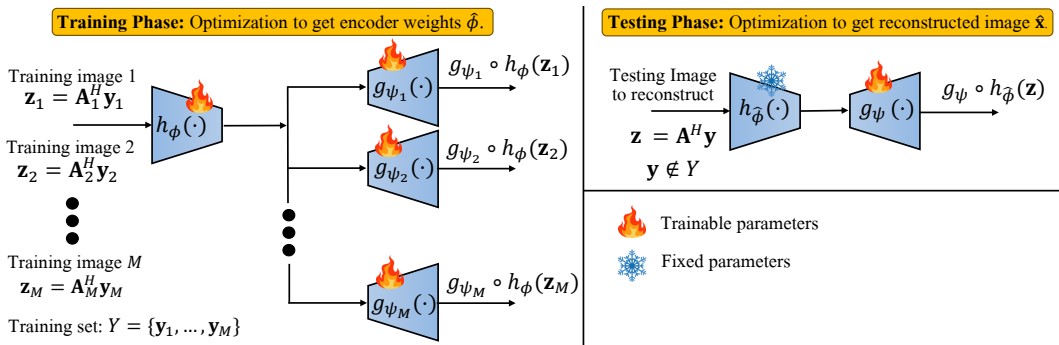

Figure 1: Illustrative block diagram UGoDIT. During training (*left*), we learn the shared encoder by optimizing over $\phi, \psi_1, \ldots, \psi_M$. At inference (*right*), given some unseen measurement vector $\mathbf{y}$ and its forward operator $\mathbf{A}$, we reconstruct the image $\hat{\mathbf{x}}$ by optimizing over $\psi$.

that while steps 2 to 4 describe updating the network parameters using a fixed-step gradient descent, we use the ADAM optimizer [47], initialized with learning rate $\beta$.

The computational requirements for training UGoDIT are determined by: (*i*) the $NK$ gradient-based parameter updates, and (*ii*) the number of function evaluations (network forward passes) necessary for updating $\mathbf{z}_i$, which is $MK$. See Figure 1 (*left*) for an illustrative block diagram of the training phase.

## 3.2 Testing phase

Provided with the trained weights of the shared encoder, i.e., parameters $\hat{\phi}$ from Algorithm 1, and unseen measurement vector $\mathbf{y}$ with its forward operator $\mathbf{A}$, at inference, we propose to use a new network mapping denoted by $h_\phi \circ g_\psi(\cdot)$, where $\phi$ is set to the trained weights $\hat{\phi}$. Then, $\psi$ is initialized using the standard way. More specifically, we set $\mathbf{z} \leftarrow \mathbf{A}^H \mathbf{y}$, $\phi \leftarrow \hat{\phi}$, and $\psi \sim \mathcal{N}(\mathbf{0}, \sigma_{\text{ini}}^2 \mathbf{I})$, and perform the following updates for $K$ iterations.

$$\psi \leftarrow \underset{\psi}{\operatorname{argmin}} \left[ \|\mathbf{A} g_\psi \circ h_{\hat{\phi}}(\mathbf{z}) - \mathbf{y}\|_2^2 + \lambda \|g_\psi \circ h_{\hat{\phi}}(\mathbf{z}) - \mathbf{z}\|_2^2 \right], \tag{5a}$$

$$\mathbf{z} \leftarrow g_\psi \circ h_{\hat{\phi}}(\mathbf{z}). \tag{5b}$$

In (5), we only optimize over $\psi$ while the parameters of the pre-trained encoder are fixed (i.e., $\hat{\phi}$)[5]. Similar to the training algorithm, for every input update (i.e., second equation in (5)), we use $N \ll K$ gradient steps for the optimization in (5).

The intuition of using the steps in (5) is the same as in aSeqDIP [26]. However, we emphasize that our goal here is different as in UGoDIT, the aim is to learn transferable weights for better and accelerated reconstructions at testing time. Algorithm 2 presents the procedure for solving (5). The training and testing phases of UGoDIT are illustrated in Figure 1.

## 3.3 Intuition behind using $M$ decoders in the training of UGoDIT

In this subsection, we describe the rationale behind using $M$ disentangled decoders in the UGoDIT training phase and how this design choice influences the quality of the learned encoder weights. Specifically, we aim to answer the following question: *Why does learning with multiple decoders result in a better shared encoder for test-time reconstruction?*

To build this intuition, we first examine the standard architecture comprising a single encoder and a single decoder (single ED)–denoted by $h_{\phi_s} \circ g_{\psi_s}(\cdot)$–trained across $M$ degraded images. The parameters are initialized as $\phi_s, \psi_s \sim \mathcal{N}(\mathbf{0}, \sigma_{\text{ini}}^2 \mathbf{I})$ and the inputs as $\mathbf{z}_i \leftarrow \mathbf{A}_i^H \mathbf{y}_i, \forall i \in [M]$. Then,

---

[5]In Appendix F, we conduct a study to show the difference between freezing the encoder (our method) and only initializing it (as was done in STRAINER [40]).

---

**Algorithm 2 Testing**: Unsupervised **G**roup **DI**P **T**ransferable via weights (`UGoDIT`-$M$)

---

**Input**: Pre-trained encoder weights $\hat{\phi}$ from Algorithm 1, measurement vector $\mathbf{y}$, forward operator $\mathbf{A}$, number of input updates $K$, number of gradient updates $N$, regularization parameter $\lambda$, and learning rate $\beta$.

  **Output**: Reconstructed image $\hat{\mathbf{x}}$.

  **Initialization**: $\mathbf{z} \leftarrow \mathbf{A}^H \mathbf{y}$ ; $\psi \sim \mathcal{N}(\mathbf{0}, \sigma_{\text{ini}} \mathbf{I})$.

1: **For** $K$ iterations, **Do**

2:     **For** $N$ iterations, **Do** (Decoder parameters update)

3:         $\psi \leftarrow \psi - \beta \nabla_\psi \Big[ \|\mathbf{A} g_\psi \circ h_{\hat{\phi}}(\mathbf{z}) - \mathbf{y}\|_2^2 + \lambda \| g_\psi \circ h_{\hat{\phi}}(\mathbf{z}) - \mathbf{z}\|_2^2 \Big]$

4:     **Obtain** $\mathbf{z} \leftarrow g_\psi \circ h_{\hat{\phi}}(\mathbf{z})$. (Network input update)

5: **Reconstructed Image:** $\hat{\mathbf{x}} \leftarrow g_\psi \circ h_{\hat{\phi}}(\mathbf{z})$

---

we perform the following alternating optimization (which extends [26] to the $M > 1$ case).

$$\phi_s, \psi_s \leftarrow \underset{\phi_s, \psi_s}{\operatorname{argmin}} \Big[ \sum_{i \in [M]} \|\mathbf{A}_i g_{\psi_s} \circ h_{\phi_s}(\mathbf{z}_i) - \mathbf{y}_i\|_2^2 + \lambda \| g_{\psi_s} \circ h_{\phi_s}(\mathbf{z}_i) - \mathbf{z}_i\|_2^2 \Big] , \tag{6a}$$

$$\mathbf{z}_i \leftarrow g_{\psi_s} \circ h_{\phi_s}(\mathbf{z}_i), \forall i \in [M] . \tag{6b}$$

Here, a single decoder $g_{\psi_s}$ is shared across all training images. While the inputs $\mathbf{z}_i$ are updated and initialized individually, they are all later optimized using the same encoder-decoder network. The gradient updates for this architecture are:

$$\psi_s \leftarrow \psi_s - \beta \nabla_{\psi_s} \Big[ \sum_{i \in [M]} \|\mathbf{A}_i g_{\psi_s} \circ h_{\phi_s}(\mathbf{z}_i) - \mathbf{y}_i\|_2^2 + \lambda \| g_{\psi_s} \circ h_{\phi_s}(\mathbf{z}_i) - \mathbf{z}_i\|_2^2 \Big] , \tag{7a}$$

$$\phi_s \leftarrow \phi_s - \beta \nabla_{\phi_s} \Big[ \sum_{i \in [M]} \|\mathbf{A}_i g_{\psi_s} \circ h_{\phi_s}(\mathbf{z}_i) - \mathbf{y}_i\|_2^2 + \lambda \| g_{\psi_s} \circ h_{\phi_s}(\mathbf{z}_i) - \mathbf{z}_i\|_2^2 \Big] . \tag{7b}$$

From these expressions, the encoder $h_{\phi_s}$ is trained with using back-propagated updates from a single decoder. In contrast, in `UGoDIT`, the encoder receives supervision from $M$ distinct decoders, each independently optimized to reconstruct a different training image (steps 3 and 4 in Algorithm 1). This over-parameterization allows each decoder to specialize in reconstructing its corresponding input, thereby improving the enforcement of measurement consistency across the training set.

Moreover, because the encoder must support multiple decoders reconstructing different training samples, it is encouraged to extract more generalizable and robust features. This leads to the intuition: *encoders that yield more accurate reconstructions during training tend to reconstruct better during testing*. We empirically support this claim in Section 4 by comparing both training and test-time reconstructions using a shared decoder versus disentangled decoders (our method).

Next, we offer a justification from the literature of multi-task representation learning. In our setting, each degraded image corresponds to an image-specific inverse problem, which can be interpreted as a separate task.

Our training setup aligns with the multi-task learning framework studied in [48], where a shared representation (encoder $h_\phi$) is learned jointly across multiple tasks, each with its own function (i.e., task-specific decoder $g_{\psi_i}$ for $i \in [M]$). Specifically, Theorem 2 in [48] demonstrates that this structure (*i*) reduces generalization error across tasks and (*ii*) promotes robust and transferable feature learning in the shared encoder. Importantly, these theoretical results rely on the structural assumption of a shared encoder with task-specific decoders. In contrast, the single ED setup (as in Eq. (6)) violates this assumption, since the decoder receives gradients from multiple tasks simultaneously. This prevents specialization and introduces gradient interference — a known challenge in multi-task learning [49]. Consequently, the theoretical benefits shown in [48] do not extend to the single ED setup which serves as an additional motivation of our proposed multi-decoder method.

# 4 Experimental results

## 4.1 Settings

We consider three tasks: MRI reconstruction from undersampled measurements, super resolution (SR), and non-linear deblurring (NDB). For MRI, we use the knee portion of the fastMRI dataset [22]. The forward model is $\mathbf{y} \approx \mathbf{A}\mathbf{x}^*$. The multi-coil data is obtained using 15 coils and is cropped to a resolution of $320 \times 320$ pixels. To simulate undersampling of the MRI k-space, we use a Cartesian sampling pattern with 4x and 8x acceleration factors (AF). Sensitivity maps for the coils are obtained using the BART toolbox [50]. For the tasks of SR and NDB, we use the FFHQ dataset [51]. For all three tasks, we test UGoDIT and baselines with 20 randomly selected degraded images[6]. Backbone architectures are adopted from [25] and [52] for MRI and SR (and NDB), respectively. For evaluation metrics, we use the Peak Signal to Noise Ratio (PSNR), the Structural SIMilarity (SSIM) index [53], and run-time. For natural images, we also report LPIPS [54]. All the experiments are run on a single RTX5000 GPU machine.

For training-data-free DIP-based methods, we use three regularization-based recent methods aSeqDIP [26], Self-Guided DIP [36], and Deep Random Projector [28] (as a DIP acceleration method), and one early stopping (ES) method, ES-DIP [37]. For data-driven MRI baselines, we use two recent DM-based methods, SITCOM-MRI [14] and DDS [12], one self-supervised method, Test Time Training (TTT) [55], and two supervised models, VarNet [18] and Recurrent VarNet [56]. For natural images, we also use SITCOM [14], in addition to DPS [29] and STRAINER [40]. STRAINER is a representative of low-data regime methods. The selection criteria of these baselines depend on methods' ability to achieve competitive reconstruction quality and high robustness to noise overfitting (for the case of DIP) for which several medical and natural image recovery tasks were considered (linear and non-linear). Further implementation details are provided in Appendix O.

For UGoDIT, we present results with three models, trained with 4, 5, and 6 degraded images (see Appendix L for results with $M \in \{2, 3, 10, 10\}$). We test UGoDIT and other data-driven methods with in-distribution test instances, i.e., images from the testing sets of fastMRI and FFHQ. Out-of-distribution evaluation is provided in Appendix A. Ablation studies on the number of layers and the number of training images are given in Appendix J and Appendix L, respectively. Step size is set to $\beta = 0.0001$. The regularization parameter is set to $\lambda = 2$, following the study in Appendix G (where we also show the robustness to noise overfitting). For $(N, K)$, we use $(2, 2000)$, $(10, 2000)$, and $(10, 2000)$ for MRI, SR, and NDB, respectively, following the study in Appendix M.

## 4.2 Impact of using $M$ decoders in UGoDIT

Here, we examine the effect of the multi-decoder architecture proposed in the training of UGoDIT. Specifically, we assess how using $M > 1$ disentangled decoders influences the quality of the shared encoder's learned weights during both training and testing for the super resolution task. To this end, we compare our method against a baseline where $M = 6$ degraded are trained using a standard single-encoder, single-decoder architecture, as described in (6). At inference, both approaches follow the procedure in (5) using their respective trained encoders.

Figure 2 shows the average PSNR curves during training (dashed) and testing (solid with 20 test images) of UGoDIT and the shared-decoder case. As seen, employing multiple decoders yields approximately over a 1 dB to 1.5 dB improvement in both training and testing. These findings empirically support the rationale discussed in Section 3.3.

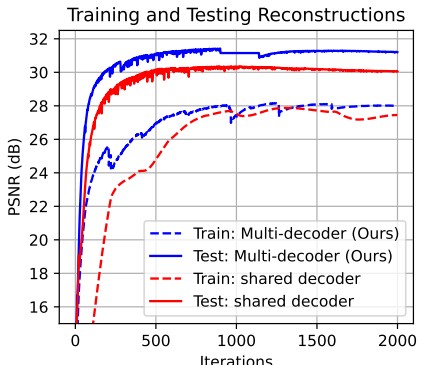

Figure 2: Averaged PSNR curves (6 for training and 20 for testing) of UGoDIT vs. the shared-decoder case (i.e., single E-D (6)) on the task of SR. See Appendix H for an additional study that highlights the benefit of the shared encoder at testing.

---

[6]In Appendix B, we show the impact of the selection of the training set on the performance of UGoDIT.

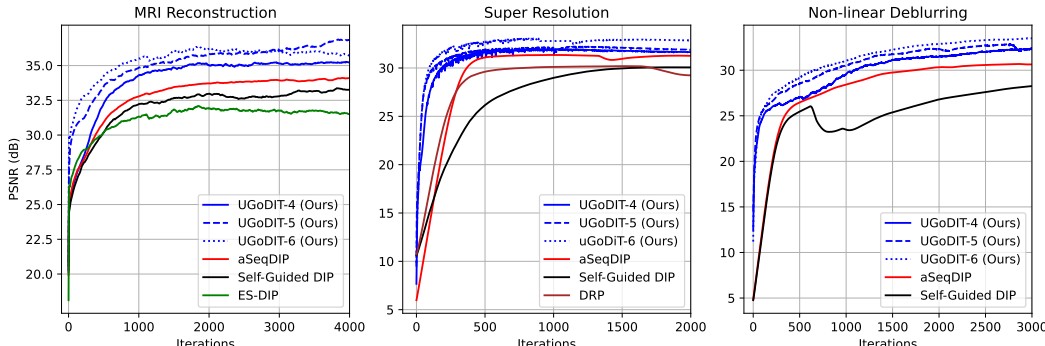

Figure 3: PSNR curves of `UGoDIT`-$M$ vs. training-data-free DIP-based baselines for the tasks of MRI (*left*), SR (*middle*), and NDB (*right*) averaged over 20 test images. Iterations (x-axis) correspond to $NK$.

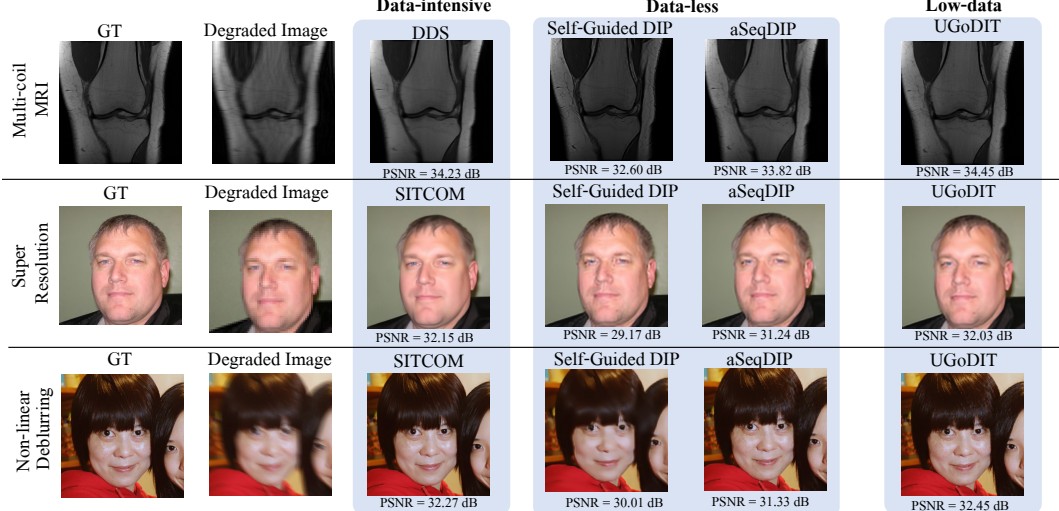

Figure 4: Reconstructed/recovered images using `UGoDIT` (last column with $M = 6$) and baselines (columns 3 to 5). As observed, `UGoDIT` return comparable PSNR results with data-intensive methods all without the need of large amount of clean images (or fully sampled measurement vectors). See Appendix P for more visualizations.

Furthermore, we observe a gap of approximately 3 to 4 dB between the training and testing curves. We hypothesize that this arises because, during training, the network parameters are jointly optimized to reconstruct $M = 6$ images, whereas the testing curve reflects the average performance over 20 separate inference-time optimizations. Appendix I provides an additional study on the learned features of `UGoDIT`.

## 4.3 Convergence of `UGoDIT` compared to training-data-free DIP-based methods

Here, we aim to answer the question: *Given a set of degraded test images, does using `UGoDIT` offer any advantages over running a DIP-based method independently on each image?*[7] Figure 3 presents the average PSNR curves, comparing `UGoDIT` (trained with 4, 5, and 6 degraded images) with four recent SOTA DIP baseline methods.

In terms of PSNR improvements, `UGoDIT` achieves at least a 2 dB gain over aSeqDIP for MRI. For SR, we observe a marginal improvement—around 1 dB or slightly less—compared to aSeqDIP, but more than 2 dB over DRP and Self-Guided DIP. For NDB, `UGoDIT` consistently outperforms the second-best method (aSeqDIP) by at least 1 dB. Regarding acceleration, `UGoDIT` demonstrates a clear advantage in all three tasks, as measured by the number of iterations required to reach target PSNR values. For instance, in MRI, `UGoDIT` reaches 32.5 dB in 500 iterations or fewer, whereas aSeqDIP requires over 750 iterations to achieve the same PSNR. This trend also holds for SR and NDB. For example, in SR, `UGoDIT`-6 requires only about 125 steps to reach 30 dB, while aSeqDIP needs

---

[7]A comparison with training-data-free measurement splitting methods is given in Appendix D.

| Method | Training images $M$ | Type | Acceleration Factor (AF): 4x | | | Acceleration Factor (AF): 8x | | |
|---|---|---|---|---|---|---|---|---|
| | | | PSNR ($\uparrow$) | SSIM ($\uparrow$) | run-time ($\downarrow$) | PSNR ($\uparrow$) | SSIM ($\uparrow$) | run-time ($\downarrow$) |
| VarNet [18] | 8000 | Fully sampled | $33.06_{\pm0.59}$ | $0.854_{\pm0.12}$ | $5.02_{\pm2.12}$ | $32.03_{\pm0.35}$ | $0.829_{\pm0.08}$ | $5.45_{\pm2.11}$ |
| Recurrent VarNet [56] | 8000 | Fully sampled | $35.88_{\pm0.46}$ | $0.958_{\pm0.08}$ | $4.82_{\pm3.34}$ | $32.33_{\pm0.15}$ | $0.879_{\pm0.12}$ | $4.55_{\pm3.21}$ |
| TTT [55] | 300 | Fully sampled | $33.81_{\pm0.45}$ | $0.881_{\pm0.12}$ | $50.81_{\pm3.14}$ | $31.83_{\pm0.24}$ | $0.879_{\pm0.15}$ | $48.25_{\pm7.31}$ |
| DDS [12] | 19460 | Fully sampled | $34.95_{\pm0.74}$ | $0.954_{\pm0.06}$ | $25.20_{\pm5.22}$ | $31.72_{\pm1.88}$ | $0.876_{\pm0.06}$ | $31.02_{\pm2.01}$ |
| SITCOM-MRI [14] | 19460 | Fully sampled | $36.33_{\pm0.37}$ | $0.962_{\pm0.08}$ | $76.24_{\pm6.25}$ | $32.78_{\pm0.64}$ | $0.892_{\pm0.02}$ | $82.45_{\pm4.20}$ |
| UGoDIT (Ours) | 4 | Sub-sampled | $35.45_{\pm0.34}$ | $0.957_{\pm0.08}$ | $45.20_{\pm4.20}$ | $32.04_{\pm0.45}$ | $0.878_{\pm0.06}$ | $47.42_{\pm5.64}$ |
| UGoDIT (Ours) | 5 | Sub-sampled | $35.74_{\pm0.41}$ | $0.959_{\pm0.09}$ | $48.12_{\pm5.42}$ | $32.36_{\pm0.51}$ | $0.881_{\pm0.06}$ | $48.12_{\pm3.24}$ |
| UGoDIT (Ours) | 6 | Sub-sampled | $36.05_{\pm0.34}$ | $0.960_{\pm0.08}$ | $48.42_{\pm5.40}$ | $32.54_{\pm0.45}$ | $0.883_{\pm0.06}$ | $49.33_{\pm7.44}$ |

Table 1: MRI reconstruction average results (with run-time in seconds) across two AFs of UGoDIT vs. data-centric methods. In SITCOM-MRI and DDS, the DM is trained on 973 MRI volumes each with 20 images. Column 3 corresponds to the "Type" of the training images. Values past $\pm$ represent the standard deviation.

| Method | Training images $M$ | Type | Super Resolution (SR) | | | | Non-linear Deblurring (NDB) | | | |
|---|---|---|---|---|---|---|---|---|---|---|
| | | | PSNR ($\uparrow$) | SSIM ($\uparrow$) | LPIPS ($\downarrow$) | run-time ($\downarrow$) | PSNR ($\uparrow$) | SSIM ($\uparrow$) | LPIPS ($\downarrow$) | run-time ($\downarrow$) |
| DPS [29] | 49000 | Clean | $24.44_{\pm0.56}$ | $0.801_{\pm0.032}$ | $0.26_{\pm0.022}$ | $75.60_{\pm15.20}$ | $23.42_{\pm2.15}$ | $0.757_{\pm0.042}$ | $0.279_{\pm0.067}$ | $93.00_{\pm26.40}$ |
| SITCOM [14] | 49000 | Clean | $30.68_{\pm1.02}$ | $0.867_{\pm0.045}$ | $0.142_{\pm0.056}$ | $27.00_{\pm4.80}$ | $30.12_{\pm0.68}$ | $0.903_{\pm0.042}$ | $0.145_{\pm0.037}$ | $33.45_{\pm9.40}$ |
| STRAINER [40] | 4 | Clean | $29.03_{\pm1.30}$ | $0.841_{\pm0.12}$ | $0.189_{\pm0.045}$ | $32.12_{\pm6.20}$ | $27.20_{\pm1.30}$ | $0.812_{\pm0.21}$ | $0.201_{\pm0.055}$ | $33.40_{\pm7.24}$ |
| STRAINER [40] | 5 | Clean | $29.24_{\pm0.89}$ | $0.846_{\pm0.15}$ | $0.181_{\pm0.056}$ | $33.45_{\pm5.78}$ | $27.39_{\pm0.86}$ | $0.817_{\pm0.24}$ | $0.197_{\pm0.035}$ | $32.60_{\pm7.24}$ |
| STRAINER [40] | 6 | Clean | $29.45_{\pm0.56}$ | $0.849_{\pm0.08}$ | $0.177_{\pm0.041}$ | $31.12_{\pm6.20}$ | $27.41_{\pm0.56}$ | $0.819_{\pm0.043}$ | $0.189_{\pm0.048}$ | $30.50_{\pm4.67}$ |
| UGoDIT (Ours) | 4 | Degraded | $30.55_{\pm0.67}$ | $0.859_{\pm0.045}$ | $0.154_{\pm0.067}$ | $29.25_{\pm5.54}$ | $29.89_{\pm0.52}$ | $0.897_{\pm0.078}$ | $0.167_{\pm0.045}$ | $28.88_{\pm4.67}$ |
| UGoDIT (Ours) | 5 | Degraded | $30.62_{\pm0.38}$ | $0.863_{\pm0.067}$ | $0.149_{\pm0.035}$ | $30.15_{\pm6.74}$ | $30.09_{\pm0.37}$ | $0.899_{\pm0.057}$ | $0.156_{\pm0.035}$ | $29.29_{\pm3.78}$ |
| UGoDIT (Ours) | 6 | Degraded | $30.75_{\pm0.58}$ | $0.871_{\pm0.054}$ | $0.157_{\pm0.058}$ | $30.41_{\pm5.54}$ | $30.19_{\pm0.46}$ | $0.906_{\pm0.067}$ | $0.147_{\pm0.035}$ | $30.18_{\pm3.89}$ |

Table 2: Restoration results of UGoDIT vs. data-centric methods. Column 3 corresponds to the "Type" of the training images. Following [29, 14], we use $\sigma_{\mathbf{y}_i} = 0.05, \forall, i \in [M]$ and $M \in \{4, 5, 6\}$. UGoDIT uses the degraded versions of the clean training images used for STRAINER. Values past $\pm$ represent standard deviation.

more than 250, and DRP nearly 600. Another noteworthy observation is that increasing the number of training images used by UGoDIT generally leads to better reconstruction quality. Collectively, these results highlight that UGoDIT provides improvements in both reconstruction performance and convergence speed. We refer the reader to Appendix E for more insights and additional results about convergence speed and training (and testing) run-times.

### 4.4 Comparison with data-centric methods

Here, we compare UGoDIT with data-driven methods. Table 1 (resp. Table 2) presents the results for MRI (resp. SR and NDB). For MRI, in terms of PSNR, all three variants of UGoDIT outperform DDS, TTT, and VarNet. Only UGoDIT-6 outperforms Recurrent VarNet (a more advanced supervised model). As expected, UGoDIT is slower than supervised methods (VarNet and Recurrent VarNet). Compared to SITCOM-MRI, UGoDIT achieves slightly lower PSNR but is faster in terms of run-time.

For SR and NDB, UGoDIT achieves comparable PSNR performance to SITCOM. Compared to STRAINER, UGoDIT provides improvements of over 1 dB for SR and 2 dB for NDB, given similar run-times—importantly, without requiring clean images for training. Overall, these results demonstrate that UGoDIT can, under the considered settings, achieve competitive performance with SOTA data-driven methods, all without relying on large quantities of fully sampled (i.e., clean) training images. See also the visualizations in Figure 4.

## 5 Conclusion

This paper introduced UGoDIT, a fully unsupervised framework for solving inverse imaging problems in the low-data regime, where only a very small number of degraded images are available during training. By learning a shared encoder with transferable weights across multiple decoders, UGoDIT effectively bridges the gap between the recent SOTA data-intensive generative models and training-data-free deep mage prior (DIP) methods. Unlike DIP approaches that optimize a separate network per image, UGoDIT leverages the shared encoder to accelerate convergence and improve reconstruction quality. Empirical evaluations across MRI, super-resolution, and non-linear deblurring tasks show that, under the considered settings, UGoDIT achieves results comparable to SOTA supervised and diffusion-based methods, all without requiring access to a large number of clean ground-truth images. These results highlight the potential of UGoDIT as a data-efficient solution for inverse problems. Limitations and future works are discussed in Appendix N.

## Acknowledgments and Disclosure of Funding

This work was supported by the National Science Foundation (NSF) grants CCF-2212065, CCF-2212066, and the CAREER Award CCF-2442240.

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

# Appendix

In the Appendix, we first present two experiments to evaluate the out-of-distribution performance of UGoDIT (Appendix A). Appendix B presents a study that shows the impact of the selection of the training set. A study on training UGoDIT with clean images is shown in Appendix C. Appendix D presents comparison results with measurement-splitting methods, followed by additional run-time experiments provided in Appendix E. Appendix F examines the impact of freezing the encoder weights at test time, followed by a study on robustness to noise overfitting (Appendix G). A study on reconstructing the test image during the training UGoDIT is given in Appendix H. Appendix I investigates the learned features in an MRI-trained UGoDIT model. This is followed by ablation studies on the number of convolutional layers (Appendix J), the number of frozen layers in the encoder of UGoDIT (Appendix K), the number of training images $M$ (Appendix L), and the choice of $N$ and $K$ (Appendix M). Limitations and future directions are discussed in Appendix N. Appendix O provides implementation details of the baselines, followed by a set of figures for additional visualizations (Appendix P).

## A  Out-of-distribution evaluation

### A.1  MRI

In this section, we investigate the out-of-distribution (OOD) capabilities of UGoDIT for the task of MRI. Specifically, we evaluate the encoder of UGoDIT-6 (which we will refer to as UGoDIT-OOD) trained on knee scans with 20 brain MRI test scans randomly selected from the fastMRI brain dataset.

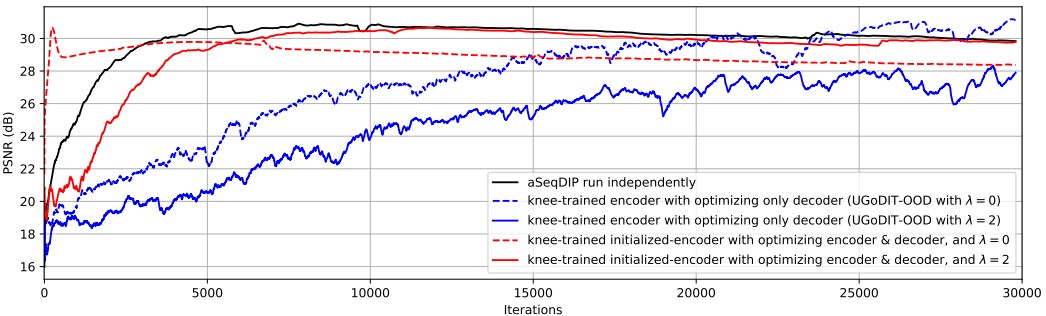

Figure 5: Average PSNR of 20 MRI brain test scans using a knee-trained (with $M = 6$) UGoDIT-OOD (blue) vs. the case where at test-time, we use the parameters of the pre-trained encoder to only initialize the test-time encoder (red). Then, we optimize over both the encoder and decoder. Running aSeqDIP independently is also included (black). Iterations in the x-axis correspond to $NK = 30000$.

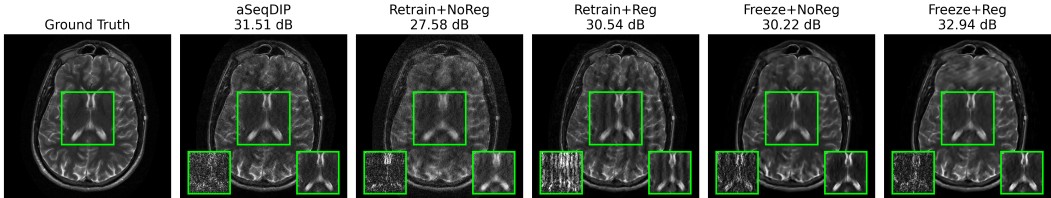

Figure 6: Brain reconstructed images for the OOD experiment. The titles of the third to fifth reconstructed images correspond to the dashed blue (UGoDIT-OOD with $\lambda = 0$), solid blue (UGoDIT-OOD with $\lambda = 2$), dashed red, and solid red curves, respectively, in the legend of Figure 5. PSNR values are given on the top of each reconstructed image.

The average PSNR curves are shown in Figure 5. As observed, when compared to aSeqDIP—which is run independently from scratch on each test scan (black curve)—the encoder trained on knee images under-performs and converges very slowly on brain scans.

We hypothesize that the autoencoding term and the input updates in UGoDIT —which implicitly enforce consistency between encoder-decoder input/output pairs—lead to the learning of features

that are overly tailored to the training distribution. As a result, when the test distribution differs significantly, these learned features make the convergence slower.

More specifically, when training and test images come from the same distribution (i.e., anatomy), the input updates and autoencoding term help reconstruction, as they promote alignment of shared low-frequency structures. However, under distribution shift (i.e., when the training and testing anatomies are different), these mechanisms degrade performance, since training and test images no longer share similar low-frequency content.

To support this hypothesis, Figure 5 includes two additional cases: (*i*) using the knee-trained encoder while optimizing only the decoder and setting $\lambda = 0$ (i.e., no autoencoding regularization in the first term of Eq. (5); shown as the dashed blue curve); and (*ii*) initializing the encoder with knee-trained weights and optimizing both the encoder and decoder, with $\lambda \in 0, 2$ (shown in red curves).

As observed, when $\lambda = 0$ (i.e., dashed curves), the results could be better than when $\lambda \neq 0$ as the input will have less impact. Furthermore, when we compare `UGoDIT`-OOD with only initializing the encoder (the red curves), the results are further improved which indicate that pre-trained weights on knees have less impact when only initialization is used. This is even more evident in the case when $\lambda = 0$ with initialization only (i.e., red dashed curve).

The visualizations in Figure 6 (which are recorded at iteration 30000) show that the all the considered cases would converge to between 27.58 dB to 31.51 dB.

## A.2 Super Resolution

In this section, we investigate the OOD capabilities of `UGoDIT` on natural images for the task of super resolution (SR). To this end, we test the encoder of `UGoDIT` (trained with $M = 6$ images from the FFHQ dataset) with 20 degraded images randomly selected from the ImageNet dataset [57].

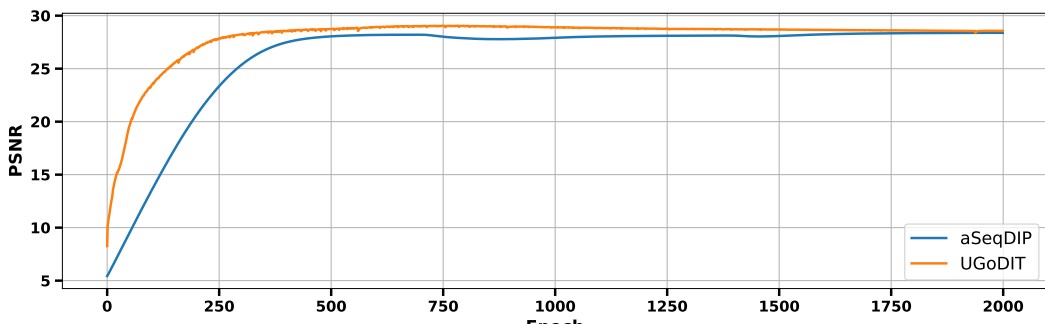

Figure 7: Average PSNR of 20 test images from the **ImageNet** dataset of `UGoDIT` trained with $M = 6$ **FFHQ** degraded images. Iterations in the x-axis correspond to $NK = 20000$.

For SR, we observe that `UGoDIT` achieves slightly better PSNR than aSeqDIP and eventually converges to similar PSNR values. However, unlike the MRI case, `UGoDIT`-OOD converges faster than aSeqDIP. This can also be observed in the restored images of Figure 8 and Figure 9. We attribute `UGoDIT` 's ability to generalize well to learning multi-frequency features under OOD scenario for the natural image.

In Table 3, we report test-time average PSNR on 20 CBSD68[8] images using two `UGoDIT`-6 models. The first model is trained using 6 images from the FFHQ dataset, and the second is trained using 6 images from the CBSD68 dataset. The results demonstrate that `UGoDIT` maintains strong performance across different datasets for the task of SR.

## B  Impact of the selection of the training set in `UGoDIT`

In this section, we provide an ablation study to demonstrate the impact of the selection of the $M$ training images for the task of SR.

---

[8] https://github.com/clausmichele/CBSD68-dataset

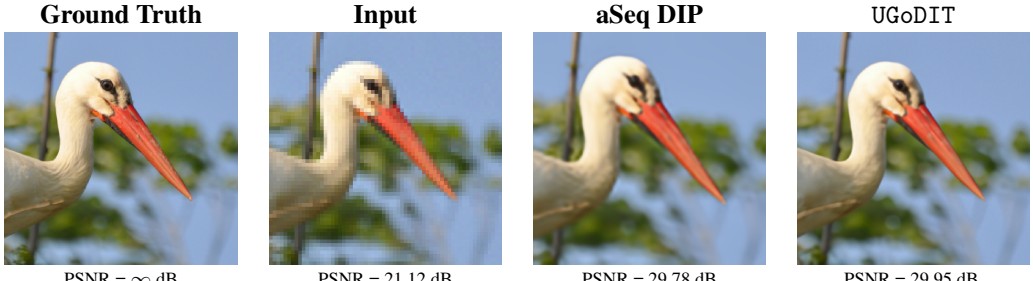

| Ground Truth | Input | aSeq DIP | UGoDIT |
|:---:|:---:|:---:|:---:|
| PSNR = $\infty$ dB | PSNR = 21.12 dB | PSNR = 29.78 dB | PSNR = 29.95 dB |

Figure 8: Ground-truth, degraded, and restored images for the OOD experiment on the task of super resolution, where UGoDIT was trained on 6 images of faces from the FFHQ dataset and tested on the ImageNet degraded image in the second column. As observed, the testing set is not a human face as is the case for the training images in FFHQ.

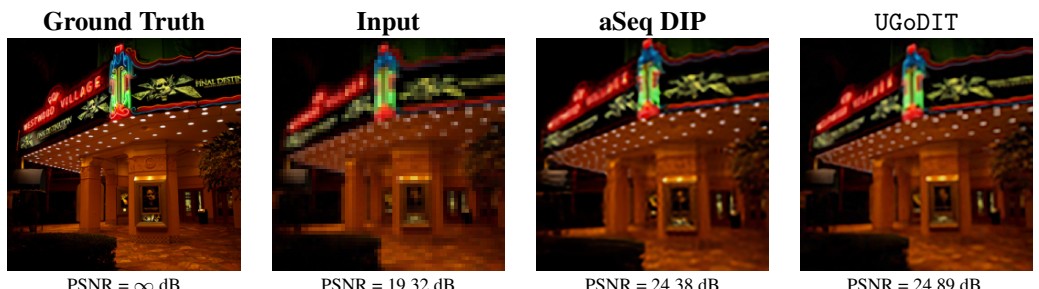

| Ground Truth | Input | aSeq DIP | UGoDIT |
|:---:|:---:|:---:|:---:|
| PSNR = $\infty$ dB | PSNR = 19.32 dB | PSNR = 24.38 dB | PSNR = 24.89 dB |

Figure 9: Ground-truth, degraded, and restored images for the OOD experiment on the task of super resolution, where UGoDIT was trained on 6 images of faces from the FFHQ dataset and tested on the ImageNet degraded image in the second column. As observed, the testing set is not a human face as is the case for the training images in FFHQ.

To this end, we evaluate six models of UGoDIT-6, each trained with a distinct set of randomly selected training images $Y_i$, i.e., for any two $i$ and $j$, we have $Y_i \cap Y_j = \emptyset$. In Table 4, we report average test-time PSNR over 100 test images.

As observed, the averaged PSNR of our method ranges between 30.58 dB to 30.95 dB, i.e., the variability using different small training sets is very small (less than 0.4 dB). This indicates that the in-distribution results of UGoDIT are not highly sensitive to the selection of training images for the task of SR.

## C Comparison of UGoDIT-$M$ vs. training on $M$ clean images

Here, we attempt to answer the question: *Does the test-time performance change if $M$ clean images are used to train the shared encoder?* Put differently, would training with $M$ clean images represent UGoDIT's best-case learning?

If clean images are available, we first need to modify the training objective since there would no longer be forward operators $\mathbf{A}_i$ or any inverse problems during training. To this end, the optimization reduces to a pure multi-decoder autoencoding loss, where the minimization in (4) becomes:

$$\phi, \psi_1, \ldots, \psi_M \leftarrow \underset{\phi, \psi_i, i \in [M]}{\operatorname{argmin}} \Big[ \sum_{i \in [M]} \| g_{\psi_i} \circ h_\phi(\mathbf{z}_i) - \mathbf{z}_i \|_2^2 \Big] , \qquad (8)$$

where $\mathbf{z}_i = \mathbf{x}_i$, for $i \in [M]$, are the clean images.

To test this idea, we compare UGoDIT trained on six degraded images vs. the multi-decoder autoencoder trained (i.e., (8)) on clean versions of UGoDIT's degraded images for the SR task.

In Table 5, we report average test-time PSNR over 100 test images.

As observed, test-time PSNR for the auto-encoder with clean data is slightly better than UGoDIT. This means that training on clean images could also serve as a prior for test-time reconstructions. The

| Method | Evaluation Type | Average PSNR over 20 CBSD68 test images |
|---|---|---|
| FFHQ-trained `UGoDIT-6` | OOD | 26.98 |
| CBSD68-trained `UGoDIT-6` | In-distribution (InD) | 27.13 |

Table 3: Average test-time PSNR on 20 CBSD68 test images using two `UGoDIT`-6 models. The first (top row) model is trained using 6 images from the FFHQ dataset (which represents the OOD case), and the second (bottom row) is trained using 6 images from the CBSD68 dataset (which represents the InD case).

| Training set for `UGoDIT-6` | Average PSNR over 100 FFHQ test images |
|---|---|
| $Y_1$ | $30.64_{\pm 0.24}$ |
| $Y_2$ | $30.57_{\pm 0.31}$ |
| $Y_3$ | $30.65_{\pm 0.27}$ |
| $Y_4$ | $30.58_{\pm 0.25}$ |
| $Y_5$ | $30.74_{\pm 0.16}$ |
| $Y_6$ | $30.95_{\pm 0.13}$ |

Table 4: Average test-time PSNR over 100 FFHQ test images for the task of SR (with $4\times$) using six `UGoDIT`-6 models, each trained with a distinct training set $Y_i, i \in [6]$.

decoder for clean training is initialized at random and then optimized to reconstruct the test image. However, we emphasize that `UGoDIT` is designed specifically for the low-data regime *without clean images*.

## D  Comparison with training-data-free self-supervised measurement-splitting methods

Here, we compare `UGoDIT`-6 with measurement-slitting training-data-free self-supervised methods: Self-supervised learning via data under-sampling (SSDU) [58] and Zero-Shot Self-Supervised Learning (ZS-SSL) [17], for the task of MRI reconstruction.

Table 6 reports the average test-time PSNR over 20 test images using two knee datasets: fastMRI [22] and the Cor-PD dataset [59].

As observed, under the considered settings, our method returns the best results as compared to these baselines.

## E  Additional insights and results on convergence speed and run-time

In this section, we provide two sets of results in regard to the convergence speed and run-time (training and testing) of `UGoDIT` as compared to the SOTA DIP-based method, aSeqDIP [26].

First, in order to highlight the convergence speed advantage of `UGoDIT` (relative to aSeqDIP), in Table 7, we report the number of optimization iterations needed to achieve 30 dB across three tasks, obtained from Figure 3. We note that, at inference, the number of parameters to optimize in `UGoDIT` is lower than aSeqDIP as the encoder in our case is fixed.

As observed, `UGoDIT` requires significantly lower number of iterations to achieve 30 dB in MRI, while requiring less than half of the number of iterations for SR and NDB.

In the second part of this section, in Table 8, we report the training time, testing time, and FLOPs for the SR task, and compare it with aSeqDIP. All methods were run for 4000 iterations.

As observed, the more training images, the higher run-time and FLOPS for `UGoDIT` as we have more decoders to train when $M$ increases. As for testing, we only require less than half the run-time as aSeqDIP since the optimization in `UGoDIT` takes place w.r.t. only the decoder, whereas aSeqDIP optimizes the entire network.

Moreover, our run-time and FLOPs are significantly lower than training diffusion models or supervised methods (data-intensive methods). For example, as per Figure 1 of [45], diffusion models require more than 17 GFLOPs to train.

| Method | Training images | Average PSNR over 100 FFHQ test images |
|---|---|---|
| `UGoDIT` | 6 degraded images | $30.75_{\pm 0.58}$ |
| Multi-decoder AE (clean training) Eq.(8) | 5 clean images | $31.02_{\pm 0.67}$ |

Table 5: Average test-time PSNR over 100 FFHQ test images of `UGoDIT`-6 model vs. a multi-decoder AE trained using (8) with the clean versions of the training images of `UGoDIT`.

| Dataset | Method | Average PSNR over 20 test slices |
|---|---|---|
| fastMRI | SSDU | 33.25 |
| | ZS-SSL | 33.47 |
| | `UGoDIT`-6 | 36.07 |
| Cor-PD | ZS-SSL | 39.55 |
| | `UGoDIT`-6 | 39.98 |

Table 6: Average test-time PSNR over 20 MRI test slices of `UGoDIT`-6 vs. measurement slitting methods using the fastMRI and Cor-PD datasets (both with 1D uniform mask).

## F   Impact of freezing the test-time encoder in `UGoDIT`

Here, we show the impact of freezing the weights of the pre-trained encoder at test time. More specifically, given a pre-trained encoder with weight $\hat{\phi}$, we compare `UGoDIT` at test time with an alternative approach that initializes the test-time encoder with $\hat{\phi}$ and then optimizes both the encoder and decoder. We note that this is a similar setting to how the pre-trained MLP weights were used in STRAINER [40].

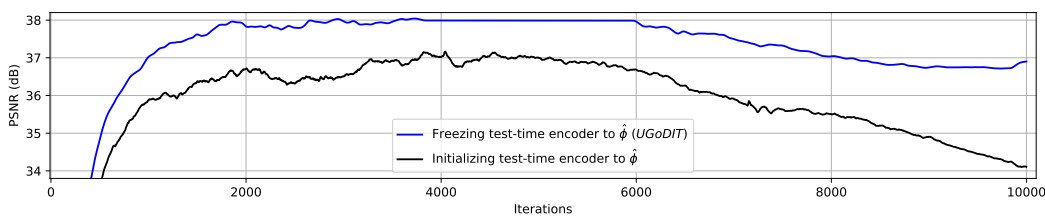

Figure 10: Average PSNR of 20 MRI test scans of `UGoDIT`-6 vs. the case where we use the parameters of the pre-trained encoder to only initialize the test-time encoder. Then, the optimization takes place over both the encoder and decoder. Iterations in the x-axis correspond to $NK = 10000$. For both cases, we use $\lambda = 2$.

Figure 10 presents the average PSNR over 20 MRI scans. As shown, freezing the encoder weights yields better results in terms of both PSNR (around 1dB difference) and convergence speed. Specifically, `UGoDIT` approaches the peak performance in approximately 2000 iterations. Moreover, `UGoDIT` requires optimizing fewer test-time parameters compared to the approach that initializes the encoder and optimizes both the encoder and decoder.

## G   Robustness to noise overfitting and the impact of the regularization parameter $\lambda$

In this section, we demonstrate how `UGoDIT` is robust to noise overfitting, typically encountered in DIP-based methods. To this end, for the task of MRI reconstruction, we train `UGoDIT` with $M = 6$ knee scans across different values of the regularization parameter: $\lambda \in \{0.1, 1, 2, 10\}$. During testing, we use 20 scans.

Figure 11 presents the average PSNR during testing. As observed, even when we run our method for 10000 iterations, the PSNR drops by approximately less than 1.5 dB. This indicates that noise overfitting takes place but is minimal given the large number of iterations.

Additionally, the case of $\lambda = 2$ returns the highest PSNR and has the least noise overfitting (as indicated by the drop in PSNR). We also observe that the difference in PSNR across different $\lambda$'s is approximately 1 dB. This indicates that the performance of `UGoDIT` is not highly sensitive to the choice of the regularization parameter. In other words, if different values of $\lambda$ are used (between 0.1 and 10), the PSNR value does not change significantly.

| Method | Number of iterations needed to achieve 30 dB | | |
|---|---|---|---|
| | MRI | SR | NDB |
| aSeqDIP | 450 | 325 | 1750 |
| UGoDIT-6 | 10 | 120 | 650 |

Table 7: Number of iterations needed to achieve 30 dB using UGoDIT-6 and aSeqDIP across the three considered tasks for the results in Figure 3.

| Method | Run-time (seconds) | GFLOPs |
|---|---|---|
| aSeqDIP | 142 | 0.01004 |
| UGoDIT-4 (training) | 384 | 0.0143 |
| UGoDIT-5 (training) | 415 | 0.0142 |
| UGoDIT-6 (training) | 472 | 0.0174 |
| UGoDIT-4 (testing) | 62 | 0.0062 |
| UGoDIT-5 (testing) | 64 | 0.0078 |
| UGoDIT-6 (testing) | 66 | 0.0084 |

Table 8: Training and testing run-time and FLOPs of different UGoDIT models.

# H  A study on reconstructing the test image during the training of UGoDIT

To further highlight the benefit of the fixed encoder (and decoder-only optimization) at testing, in this section, we report the results of Table 9 (with $M = 5$) where, given the same $M + 1$ FFHQ images, we consider:

1. The PSNR of image $M + 1$ using a pre-trained UGoDIT-$M$ encoder trained on image 1 to image $M$ (first row). In other words, the PSNR of image $M + 1$ using our test-time optimization in (5) (i.e., Algorithm 2).

2. The PSNR of image $M + 1$ when training UGoDIT-$M + 1$ jointly on all $M + 1$ images (second row). In other words, the PSNR at decoder $M + 1$ using our training optimization in (4) (i.e., Algorithm 1).

| Method | PSNR for image $M + 1$ |
|---|---|
| Pre-trained UGoDIT-$M$ encoder with test-time decoder | 29.12 |
| UGoDIT-$M + 1$ training | 25.41 |

Table 9: PSNR for image $M + 1$ for the two cases in Appendix H.

As observed, training on images 1 to $M$ and then testing on image $M + 1$ returns a better PSNR. Our interpretation of the result is as follows. When the encoder is trained on $M + 1$ images with multiple image-specific decoders, the optimization of the decoder for the $(M + 1)$-th image is affected by the encoder having to fit over all images, leading to an unfavorable convergence trade-off compared to learning the shared encoder and fixing it for the $(M + 1)$-th image and optimizing the decoder freely from random initialization. In this case, the decoder adapts more freely for the $(M + 1)$-th image without having to account for other images leading to better performance.

# I  A study on the learned features in UGoDIT

To better understand the representations learned by our shared encoder, we examine the feature maps from its first few convolutional layers across two different approaches - one with the proposed UGoDIT framework and the other with the shared decoder (termed single E-D in Section 3.3) framework.

To analyze the frequency content of the learned features, for the task of MRI, we applied a 2D Fourier transform to each channel of the encoder's layer and visualized the corresponding magnitude spectra. We computed a low-frequency (LF) magnitude ratio, which is the proportion of spectral energy contained within a central region of the frequency domain. This helps us to quantify, to how much extent, low frequency features dominate the learned representation. The LF magnitude ratio can be

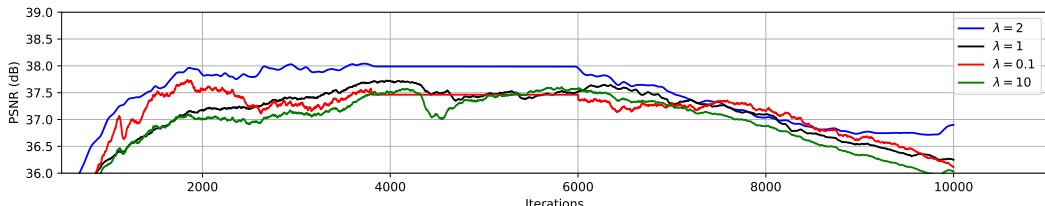

Figure 11: Average PSNR of 20 MRI test scans of `UGoDIT`-6 using different values of the regularization parameter $\lambda$ at training and testing. Iterations in the x-axis correspond to $NK = 10000$.

defined as

$$\text{LF ratio} = \frac{\sum_c \sum_{(u,v) \in \mathcal{C}} |\hat{x}_c[u,v]|}{\sum_c \sum_{(u,v)} |\hat{x}_c[u,v]|} \, , \tag{9}$$

where $\hat{x}_c[u,v]$ is the 2D Fourier transform of the feature map at channel $c$, and $\mathcal{C}$ is a centered square region in frequency space corresponding to low frequencies.

| Metric | Slice 1 | | Slice 2 | |
|---|---|---|---|---|
| | **UGoDIT** | **Single E-D** | **UGoDIT** | **Single E-D** |
| LF magnitude ratio: First Layer | 0.463 | **0.525** | 0.363 | **0.417** |
| LF magnitude ratio: Second Layer | **0.341** | 0.318 | 0.302 | **0.325** |
| LF magnitude ratio: Third Layer | 0.235 | **0.337** | 0.233 | **0.357** |
| LF magnitude ratio: Fourth Layer | 0.166 | **0.232** | 0.207 | **0.303** |
| LF magnitude ratio: Decoder Output | 0.475 | **0.500** | 0.503 | **0.534** |
| PSNR (dB) | **36.61** | 33.98 | **35.92** | 34.82 |

Table 10: Low-frequency (LF) magnitude ratios at the output of four encoder layers (first four rows) and the decoder output (fifth row), and the overall PSNR for `UGoDIT` and the Single E-D approach (i.e., (6)). The task is MRI reconstruction at $4\times$ acceleration factor. For reference, the LF magnitude ratios for the ground truth images are 0.412 (Slice 1) and 0.439 (Slice 2).

Table 10 summarizes the low-frequency (LF) magnitude ratios computed for two images across four encoder layers and the decoder output, along with PSNR (in dB). These results are reported for both the `UGoDIT` and Single E-D (one encoder, one decoder) approaches where we used an acceleration factor of $4\times$.

Across both test slices, Single E-D exhibits higher LF magnitude ratios in most layers, including the decoder output, indicating a stronger low-frequency bias. `UGoDIT` achieves higher PSNR values for both slices that have only moderate low-frequency content. Its feature representations better capture high-frequency details in the image and yield better final reconstructed images than the single E-D case, which shows a more pronounced low-frequency bias from shared training.

## J   Impact of the number of convolutional layers in the architecture of `UGoDIT`

In this section, we analyze the impact of varying the depth of the convolutional blocks in `UGoDIT`. Specifically, we evaluate five different configurations of the Skip-net architecture [52] on the super-resolution task performance, each with a different number of upsampling and downsampling convolutional layers, and report the resulting PSNR values.

As shown in Figure 12, the configuration with five convolutional layers in both the encoder and decoder produces the highest and most stable PSNR. Increasing the number of layers beyond five yields no further improvement, while reducing the layer count leads to a noticeable degradation in PSNR.

## K   Impact of the number of frozen layers in the encoder of `UGoDIT`

In this subsection, we attempt to answer the question: *How much of the encoder arm needs to be trained over the $M$ images?*

To this end, in Table 11, we report the results of `UGoDIT`-6 (MRI task) with freezing different number of encoder layers. As observed, the default setting (5 frozen layers) returns the best results.

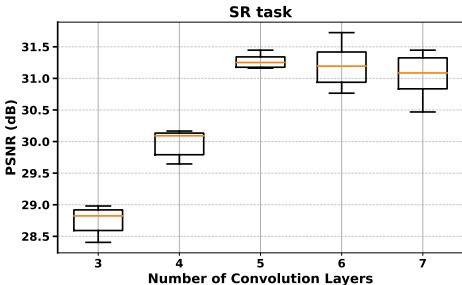

Figure 12: Ablation study in terms of PSNR as a function of the number of Convolution layers in the Skip Net for the Super Resolution task, averaged over 20 test images. The x-axis indicates the number of convolutional layers in each of the upsampling and downsampling parts of the network architecture.

| Number of encoder frozen layers at test time | Average PSNR over 20 test slices |
|:---:|:---:|
| 5 (default) | $35.89_{\pm 0.28}$ |
| 4 | $35.70_{\pm 0.43}$ |
| 3 | $35.37_{\pm 0.51}$ |
| 2 | $35.23_{\pm 0.44}$ |
| 1 | $34.89_{\pm 0.38}$ |

Table 11: Average test-time PSNR over 20 fastMRI test slices for `UGoDIT`-6 with freezing different numbers of layers in the 5-layer encoder.

## L  Impact of the number of training images ($M$) in `UGoDIT`

In this section, we investigate how the size of the training set influences `UGoDIT` 's performance. To this end, we consider the non-linear deblurring task and train the shared encoder using five different FFHQ-derived subsets [51], and evaluate each on the same deblurring benchmark. As shown in Figure 13, performance steadily improves as we increase the number of training images, peaking at six images—beyond which additional data yields negligible gains.

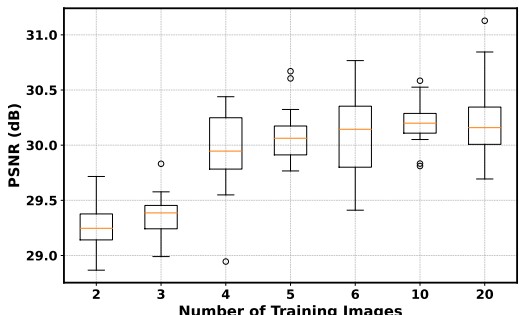

Figure 13:  Ablation study on the number of training images, $M$, as given in Algorithm 1 and Algorithm 2. Results are averaged over 20 images for the task of NDB.

## M  Impact of the selection of $N$ and $K$ in `UGoDIT`

Here, we conduct an ablation study on the number of gradient updates, $N$, for each inputs updates (i.e., $K$) in the training (Algorithm 1) and testing (Algorithm 2) of `UGoDIT`.

The results are given presented in Figure 14, where we report the average PSNR values across the tasks of MRI (considering combinations of $(N, K)$ as $(1, 4000)$, $(2, 2000)$, and $(4, 1000)$), SP, and NDB (considering combinations of $(N, K)$ as $(4, 5000)$, $(10, 2000)$, and $(20, 1000)$).

The results show that the combination we selected (for each task), on average, returns the best results.

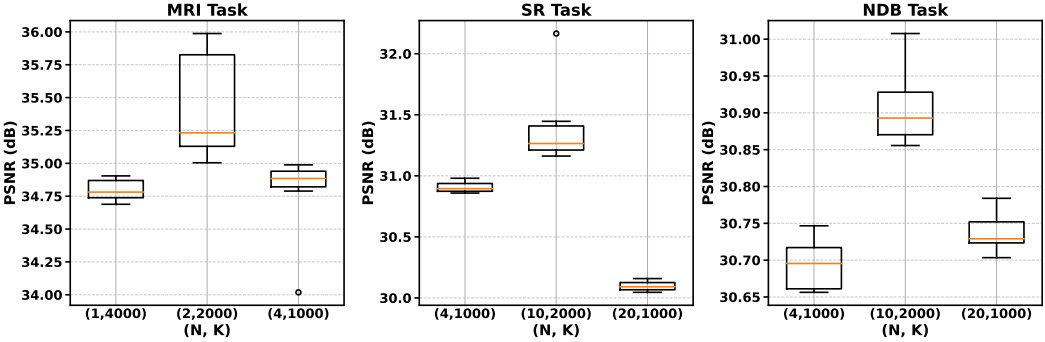

Figure 14: Ablation study on the number of gradient updates, $N$, for every inputs updates, $K$, as given in Algorithm 1 and Algorithm 2. Results are given as the average of 20 images for the tasks of MRI (*left*), SR (*middle*), and NDB (*right*).

## N    Limitations and future work

In this paper, we assume full access to the forward model and focus on 2D image recovery problems. In future work, we aim to extend UGoDIT to 3D image reconstruction and to the blind setting, where the goal is to jointly reconstruct the image and estimate the forward operator.

Additionally, our out-of-distribution (OOD) performance varies across tasks, as discussed in Appendix A. Specifically, for MRI, our OOD performance is lower than that of independently running aSeqDIP. In contrast, for super-resolution (SR), we observe improved acceleration while slightly outperforming aSeqDIP in terms of PSNR. As such, for future work, we will explore combining our method with test-time adaptation approaches to enhance performance on OOD test cases. Furthermore, we will consider cases where the training and/or testing data come from different forward models, i.e., train-test distribution mismatch.

## O    Implementation details for baselines

**Natural image restoration baselines:**    For Diffusion-based approaches, we use the authors' official code for each method: DPS with the DDPM solver using $1,000$ diffusion steps; SITCOM with $N = 20$ diffusion steps where $K = 20$ optimization steps are used for each. These methods use the pre-trained DM (trained on FFHQ) from DPS. For STRAINER, the Shared INR architecture consists of a single shared encoder—composed of six sine-activated layers—and $k$ decoder heads, where $k$ is the number of images to be learned. During training, we jointly train the encoder and all decoders by minimizing the sum of per-decoder mean squared error (MSE) losses using ADAM [47] with a learning rate of $0.001$. At test time, we instantiate a new single-decoder INR initialized with the pre-trained encoder weights and train only both using a data-consistency loss tailored to the relevant forward operator.

**MRI reconstruction baselines:**    For DM-based methods, the pre-trained model is from DDS [12]. In DDS, we set $N = 100$ diffusion steps, whereas SITCOM-MRI uses $N = 50$ diffusion steps, $K = 10$ optimization steps, and $\lambda = 0$ (no regularization). For the supervised learning baselines (i.e., VarNet and Recurrent VarNet), we adopt the default VarNet configuration, comprising 12 cascades of unrolled reconstruction steps. Optimization is performed with Adam (learning rate $3 \times 10^{-4}$) on the FastMRI knee dataset [22], which contains 973 volumes. From each volume, we discard the first and last five slices, then randomly sample 8,000 images from the remaining slices to form the training set.

## P    Visualizations

Figure 15 shows the clean versions of 6 training images (top row) and the 20 testing images sampled from the FFHQ dataset that we used in UGoDIT. Figures 16 and 17 present additional MRI visualizations. Samples from the natural image restoration tasks are given in Figure 18 and Figure 19 super resolution and non-linear deblurring, respectively.

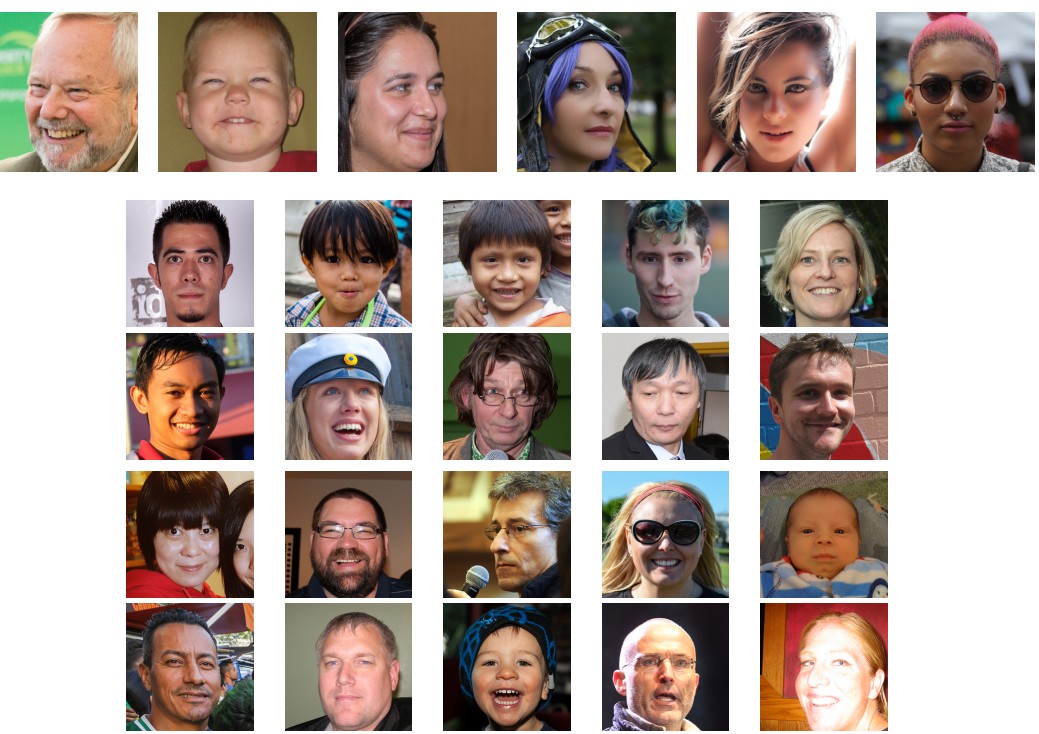

Figure 15: Top: the 6 training images. Bottom: the 20 test images.

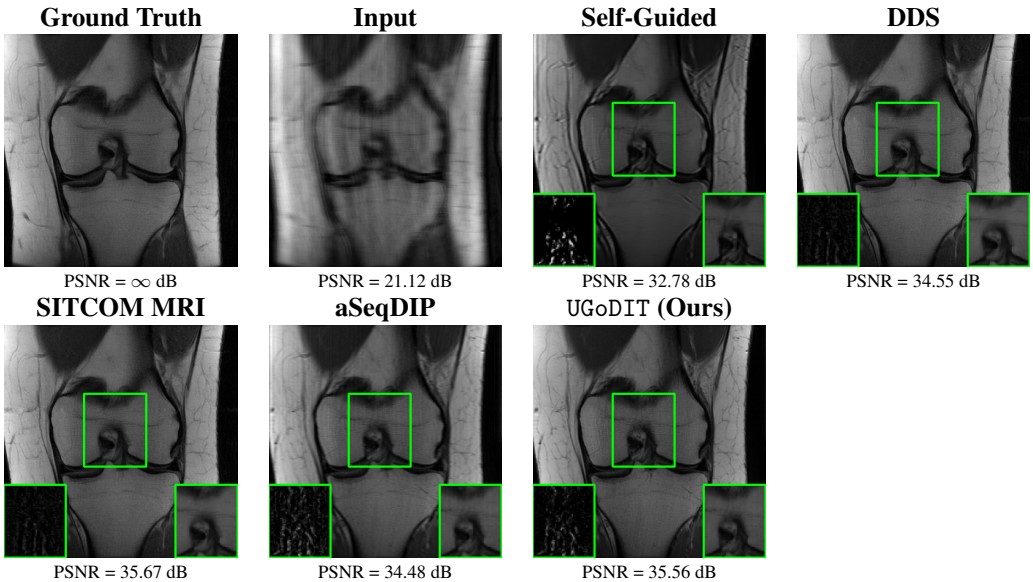

Figure 16: Visualization of ground-truth and reconstructed images using different methods of a knee image from the fastMRI dataset with 4x k-space undersampling. A region of interest is shown with a green box and its error (magnitude) is shown in the panel on the bottom left.

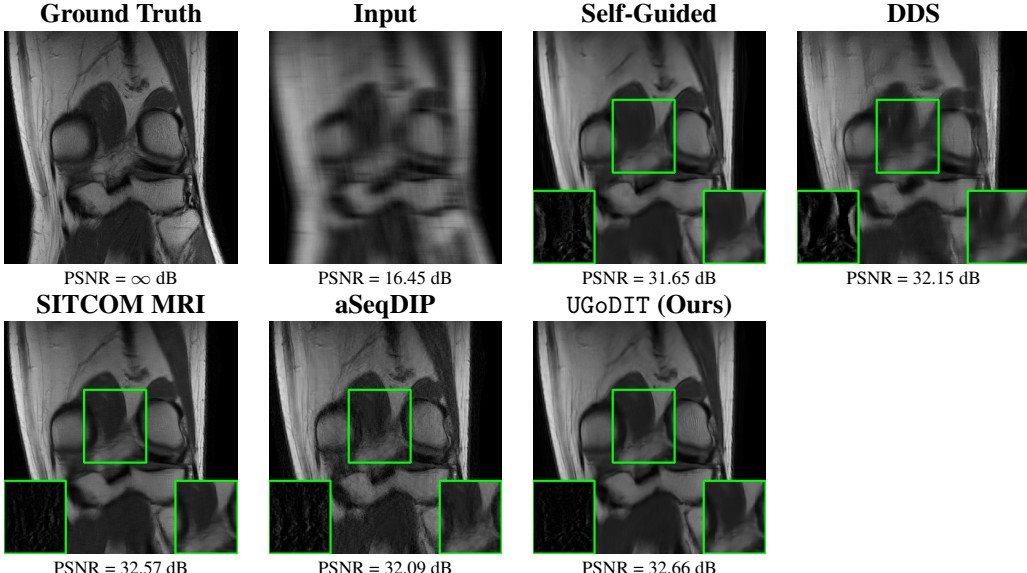

Figure 17: Visualization of ground-truth and reconstructed images using different methods of a knee image from the fastMRI dataset with 8x k-space undersampling. A region of interest is shown with a green box and its error (magnitude) is shown in the panel on the bottom left.

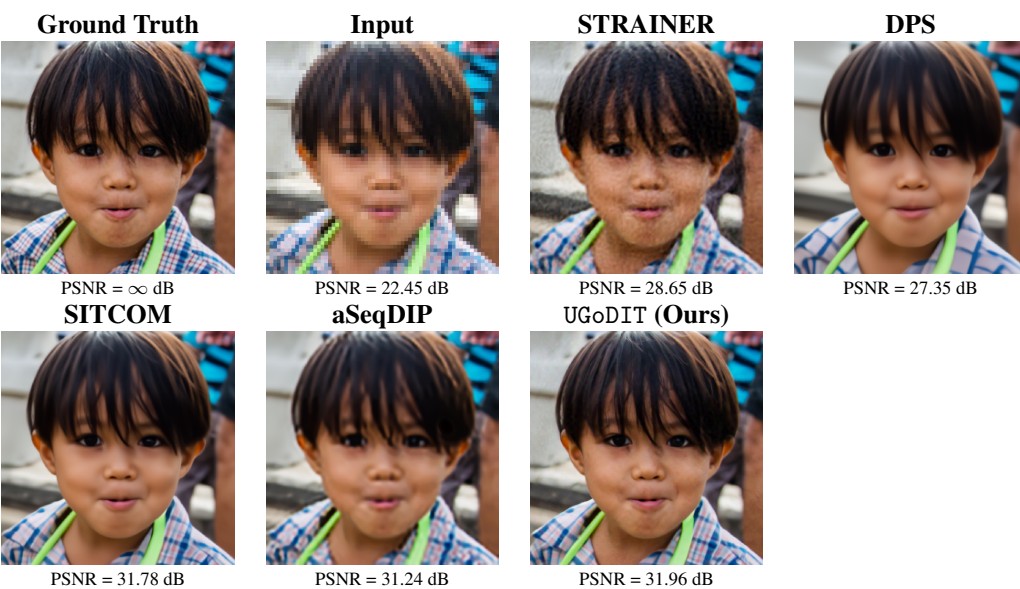

Figure 18: Super Resolution example from the FFHQ dataset.

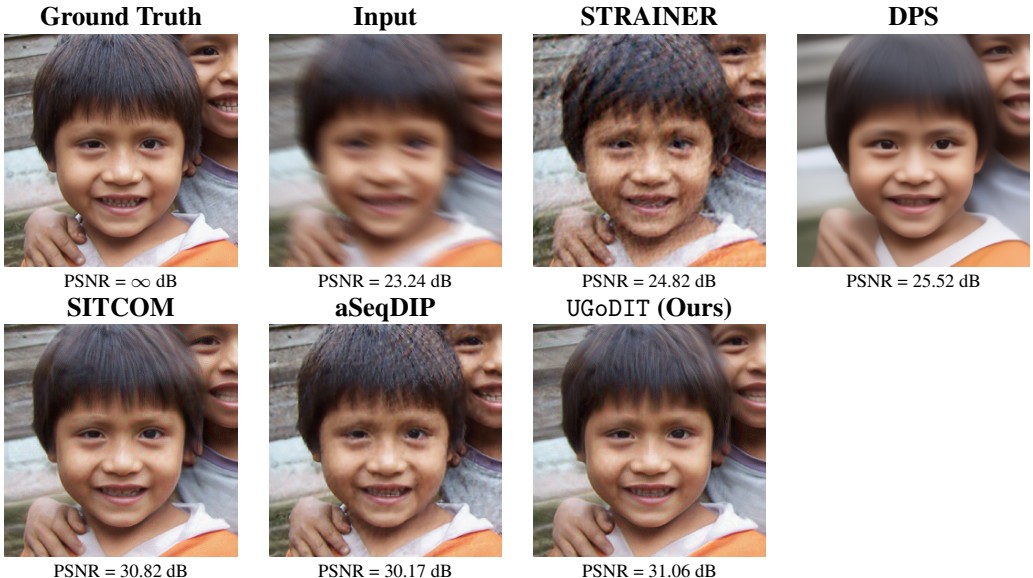

| Ground Truth | Input | STRAINER | DPS |
|---|---|---|---|
| PSNR = ∞ dB | PSNR = 23.24 dB | PSNR = 24.82 dB | PSNR = 25.52 dB |

| SITCOM | aSeqDIP | UGoDIT (Ours) |
|---|---|---|
| PSNR = 30.82 dB | PSNR = 30.17 dB | PSNR = 31.06 dB |

Figure 19: Non-linear deblurring example from the FFHQ dataset.

