# OpenReview forum: "UGoDIT: Unsupervised Group Deep Image Prior Via Transferable Weights"
_NeurIPS.cc/2025/Conference — NeurIPS 2025 poster_

### Official Review · Reviewer_oaHC · 2025-06-30

**Clarity:** 3
**Significance:** 2
**Originality:** 3
**Rating:** 3
**Confidence:** 5

**Summary:**

This paper proposes a new training approach for DIP methods in solving inverse imaging problems. Instead of the standard DIP approach, which trains a full network for a dataset, this paper concentrates on CNNs that have encoder-decoder structures, and shares the encoder for training over few datasets, while tuning the decoder individually for each problem. Additionally, this training is built into an alternating optimization algorithm that also updates the input to the network. The results show competitive results. However, there are concerns about the justification of the method, the limited evaluations and implementation details.

**Questions:**

- Why does the encoder trained only on 4-5 images generalize to unseen data well? If it does not, as eluded to in Appendix A, then what is the advantage over standard "dataless" methods
- Why are the methods not evaluated over the full test datasets?
- What are the details of the forward operators?
- What are the details of how the M images for the proposed method are selected with respect to the test cases?

**Ethical Concerns:**

["NO or VERY MINOR ethics concerns only"]

**Final Justification:**

The rebuttal has answered some of the questions raised. There are still some fundamental issues, such as the lengthy run time making these impractical or the setup where this is useful, which are inherent to the study design. I have increased my initial score accordingly.

**Limitations:**

There is a limitation section in the appendix, but this is more about future work than actual limitations, e.g. the limited evaluation and other issues are not discussed.

**Quality:**

2

**Strengths And Weaknesses:**

Strengths:
- The idea of sharing the encoder of a U-Net is reasonable, which has been explored in other scenarios before.
- The reported results show improvement over existing datasets.

Weaknesses:
- Unclear why this setup is valuable, since the run time is still quite high. There are also other non-DIP methods for test-time training that are not even discussed.
- While the idea of the shared encoder is intuitive, it is unclear why this would work in the limited data regime in the first place. This is somewhat discussed in Sec. 3.2, where it is claims that "encoders that yield more accurate reconstructions during training tend to reconstruct better during testing". However, the "more accurate" nature of the encoder is never evaluated, except via the reconstruction. Furthermore, it is not clear why the encoder trained on 5-6 images is not overfitted, and can provide reasonable representations for new test data.
- 20 test cases is not enough to evaluate these methods.
- Forward operators are not clearly defined for any of the experiments. For instance, the MRI data is most likely undersampled randomly, but this is not mentioned. It is also unclear if it will work well for uniform undersampling that typically arises in brain imaging, since 4b inherently relies on denoising of the input via projections, which applies to random undersampling but not uniform.
- It is not clear how the additional M images for the proposed methods are selected with respect to the test cases. For instance, if all MRI datasets are chosen from similar cases for this (e.g. same field strength, same contrast/SNR), this is an unfair advantage compared to the database-trained DM and supervised methods that are trained across varying conditions.
- There is something wrong with the VarNet implementation (which incidentally is outdated). The metrics are below previously published results. The run time is also excessively long, since these type of networks usually take on the order of hundred ms, not 5 seconds. It also looks like sensitivity maps were pre-generated via BART, which should further cut down on inference time. No images are shown for VarNet either.
- Similarly, PSNR/SSIM calculations on FFHQ does not seem to align with the images, e.g. in Fig. 18. In this figure, STRAINER and aSEQDIP are highly noisy/pixellated, yet outperform DPS by 1 and 4 dB respectively.
- No ablation studies on the impact of the neural network architecture choice (Appendix E does not address this) or how much of the encoder arm needs to be trained over the M images.

- Some claims in the introduction are not aligned with the literature. For instance, there are many work that train DMs for MRI/CT from scratch without adapting from natural images, or the claim that unsupervised methods fall short of generative models need citation.
- Unlike what is stated in Sec. 3.1, not all CNNs include a downsampler and an unsampler. Please rewrite this for the U-Net architecture, which is the only one considered.
- Impact of using M decoders is shown with eq. (5), why not (1)?

---

> ### Author Rebuttal · Authors · 2025-07-31
>
> We'd like to thank the reviewer for their comments. We are glad that the reviewer acknowledges our use of a multi-decoder UNet and the improvements over baselines.
>
> ## (W1) Setup value. Run-time. Comparison with test-time training methods.
>
> **Value of the setup** is in **applications where only a few degraded measurements are available** (e.g., in some scientific and medical applications). In these low-data regimes, data-centric methods requiring large clean datasets are not applicable, and our approach provides a strong case under such scenario.
>
> **Our Run time** when compared to standalone DIP-based shows that UGoDIT achieves **significantly faster convergence** because the learned encoder provides a strong initialization at test time. **See our response to Reviewer usw1 (D2) for the number of iterations needed to achieve 30dB at inference across the three tasks**.
>
> **Comparison with test-time training methods:** See our response to Reviewer eMg2 (W4+W5) for comparison with a **TTT method from ICML 2022** in addition to 2 other baselines. **We would also appreciate any specific TTT papers that the reviewer considers relevant.**
>
> ## (W2) Why the shared encoder works in the limited-data regime? The "more accurate" nature of the encoder is only evaluated by reconstruction. Why the encoder trained on 5-6 images is not overfitted?
>
> We thank the reviewer for raising these important questions.
>
> **Shared encoder in the limited-data regime:** UGoDIT jointly optimizes the encoder with multiple decoders across a small set of degraded images, which encourages it to capture **task-common features**. This inductive bias, supported by multi-task learning (see our response to Reviewer 2CSh (W1)), enables the encoder to generalize even when trained with as few as 5–6 images.
>
> **Evaluation of "encoder accuracy":** We use reconstruction quality as the primary evaluation metric because the encoder is trained solely to produce features that aid reconstruction. Thus, improved reconstruction accuracy on unseen test images directly supports the claim that the encoder has learned meaningful representations. **We would welcome suggestions from the reviewer on alternative evaluation criteria/metrics they would find more convincing**.
>
> **Encoder's features not over-fitting:** The shared encoder saves parameters and hence prevents over-fitting. In **Appendix D**, we provide a feature analysis showing that the encoder trained with multiple decoders learns smoother and more structured intermediate representations compared to a single-decoder baseline. These features transfer well to new test data, which explains the strong generalization despite the small training set.
>
> ## (W3+Q2) Evaluation on $>20$ test images.
>
> Please see our response to Reviewer 2CSh (W2+W4+Q2), where we conducted experiments **using 100 test images**.
>
> We report the results of an FFHQ-trained-UGoDIT-6 using **100 ImageNet test images** and **20 CBSD65 test images** for the SR task in response to Reviewer usw1 (Q3).
>
> ## (W4+Q3) UGoDIT MRI with uniform sampling.
>
> In our experiments, we used **variable density random sampling masks commonly used in compressed sensing MRI** [A].
>
> In response to the reviewer's suggestion, **we report the performance of UGoDIT-6 using uniform (equispaced) undersampling patterns on fastMRI brain data**. For each acceleration factor (AF), one-third of the k-space lines are fixed at the center, while the remaining lines were uniformly distributed across the high-frequency regions.
>
> | Method|PSNR at AF = 4x|PSNR at AF = 8x|
> |-|-|-|
> |UGoDIT-6|35.88 ± 0.58|32.78 ± 0.68|
> |SITCOM|36.33 ± 0.37|32.92± 0.64|
>
> As observed, UGoDIT results are still competitive using a different undersampling patterns even when compared to SITCOIM which was trained on thousands of fully sampled images.
>
> [A] Sparse MRI: The application of compressed sensing for rapid MR imaging, MRM 2007
>
> ## (W5+Q4) Selection of $M$ images. Unfair advantage (compared to high-data methods that are trained across varying conditions) if all MRI data are chosen from similar cases.
>
> We thank the reviewer for this comment. **The training and testing images were selected at random** including MRI, i.e., they are not selected to be close to the test images.
>
> In our evaluation, DMs are trained on fully sampled measurements, whereas supervised methods are trained separately for each setting. For example, in Table 1, we have two pre-trained VarNets, one for 4x and one for 8x, where 8000 pairs of sub-sampled and full-sampled images are used. In UGoDIT's main results, at most, we train on six sub-sampled images.
>
> To show the impact of the selected training images, **see to our response to Reviewer 2CSh (W2+W4+Q2)**.
>
> ## (W6) VarNet (being outdated) selection, results, run-time, and visualizations.
>
> We thank the reviewer for this comment.
>
> **On VarNet being "outdated":** We respectfully disagree with the characterization of VarNet as outdated. VarNet [17] is a paper from MICCAI 2020 (a well-known venue for MRI) that remains a widely used baseline in MRI reconstruction. However, to address the reviewer's concern, we compare with **Recurrent VarNet (CVPR 2022)** [B].
>
> | Method|PSNR|
> |-|-|
> |UGoDIT-6| 36.05 $\pm$ 0.78 |
> |Recurrent VarNet [B]|35.88 $\pm$ 0.46|
>
> Our VarNet implementation follows the official codebase but was trained on **8,000 images** (as mentioned in Table 1), which is fewer than what typically used to report SOTA results. This difference likely explains the lower metrics we observed. However, the performance of our approach trained in very low data regime is still competitive with methods trained on orders of magnitude more images indicating the advantages of expertly exploiting shared architectures, regularization, and implicit biases of neural networks to direct data-consistent reconstructions.
>
> The ~5 seconds reflects **end-to-end inference** including BART-based sensitivity map generation and data loading. This also depends on the compute power.
>
> We will include VarNet and Recurrent VarNet visuals with error maps in the revised paper. **The key message for our comparison with supervised methods is to show that UGoDIT can achieve competitive results without needing large number of paired training images.**
>
> [B] Recurrent Variational Network: A Deep Learning Inverse Problem Solver applied to the task of Accelerated MRI Reconstruction, CVPR 2022
>
> ## (W7) STRAINER and aSeqDIP are better than DPS but noisier.
>
> The reviewer is indeed correct. DPS is known to produce artifact-free but more blurred images that could often violate measurement consistency, leading to lower PSNR/SSIM despite perceptually cleaner results (see also the visualizations in Figure 5 of aSeqDIP). At every reverse step of DPS, one gradient is taken w.r.t. $y$ and $A$. In contrast, STRAINER and aSeqDIP enforce data consistency more explicitly.
>
>
> ## (W8) Ablations: (1) Impact of the NN architecture choice beyond Appendix E; (2) How much of the encoder arm needs to be trained over the $M$ images.
>
> Since our method builds on DIP, we use a standard U-Net with residual connections. If the reviewer refers to non-CNN architectures, we note that Transformers and MLPs, unlike CNNs, lack spectral bias [15]. **We kindly ask the reviewer to indicate if they refer to other CNNs such as U-Nets that don't use residual connections.**
>
> To address the second part, we report the results of UGoDIT-6 (MRI task) with freezing different number of encoder layers.
>
> |Number of encoder frozen layers at testing |PSNR|
> |-|-|
> |5(default)|35.89 $\pm$ 0.28|
> |4|35.70 $\pm$ 0.43|
> |3|35.37 $\pm$ 0.51|
> |2|35.23 $\pm$ 0.44|
> |1|34.89 $\pm$ 0.38|
>
> As observed, the default setting (5 frozen layers) return the best results. We thank the reviewer for this suggestion. We will include this study in the revised paper.
>
> ## (W9) Works that train DMs from scratch without adapting from natural images; Unsupervised methods fall short of generative models need citation.
>
> We say that DM-based methods **often** use fine-tuning in MRI/CT which is the case in several recent methods [12,13,17]. **Our sentence does not rule out the existence of other methods that train MRI/CT DMs from scratch. We would greatly appreciate it if the reviewer could point us to such works.**
>
> Thanks for suggesting to cite paper(s) that supports the claim: unsupervised methods **often** fall behind DM methods. This claim is supported in works such as [C] as well as the results of baseline SITCOM in Tables 1 and 2, where a DM method is shown to be better than our proposed unsupervised method.
>
> [C] Sequential Diffusion-Guided Deep Image Prior for Medical Image Reconstruction, ICASSP 2025
>
> ## (W10) Description of UNet in Sec. 3.1.
>
> We will re-write according to the reviewer's suggestion.
>
> ## (W11) Impact of using $M$ decoders is shown with (5), why not (1)?
>
> Since Eq. (1) corresponds to vanilla DIP, overfitting is expected. Eq. (5) describes the single-encoder/single-decoder case, which—similar to UGoDIT—uses the autoencoding term with input updates that help mitigate overfitting.
>
> Moreover, there is no distinction between training and testing for Eq. (1), as vanilla DIP is a training-data-free method and does not involve $M$ training images.
>
> ## (Q1) Advantage over dataless methods if UGoDIT does not generalize well on OOD.
>
> In Appendix A.1, our method with MRI does indeed require a large number of iterations to converge. However, SR results in Appendix A.2 show that UGoDIT generalizes very well to OOD data in terms convergence speed and PSNR. Therefore, for MRI, if faced with unseen data that is from a different anatomy, using UGoDIT or aSeqDIP would return similar PSNRs (see Figure 5), but for fast convergence, it would be better to use a dataless method, such as aSeqDIP.
> We thank the reviewer for their comment. **See our response to Reviewer 2CSh (W1) for generalization. For an analysis on the learned features, see Appendix D.**

---

> > ### Comment · Reviewer_oaHC · 2025-08-05
> >
> > I thank the authors for their responses.
> >
> > While I can see their method improving on DPS, which approximates posterior sampling, I am still having a hard time understanding how and why their method can outperform well-trained supervised methods matched to the test setup. The explanations provided here do not clarify this convincingly.
> >
> > Second, as I raised the concern with VarNet performance initially, the added experiments add to the confusion. Looking at the data provided in the rebuttal here (which incidentally also does not clearly define the forward operators) on Recurrent VarNet and the data provided in the rebuttal for Reviewer eMg2, the reported performance of the comparison methods are up to 7 dB lower than what is reported in the respective papers. It would be hard to explain such a huge difference with just a random selection of evaluation points, so I'm still wondering if there are any implementation issues here.

---

> > > ### Author Response · Authors · 2025-08-06
> > >
> > > We'd like to thank the reviewer for engaging in the discussion.
> > >
> > > ## Further explanation on the supervised methods results.
> > >
> > > We thank the reviewer for his comment. For VarNet and Recurrent VarNet, we train the default architecture with 8000 training images and 12 cascades (unrolling steps) using the Adam optimizer with 0.0003 learning rate. The number of training images used in VarNet and Recurrent VarNet papers are higher. This means that there could be a Recurrent VanNet model (trained on >> 8000 training images) that could return results that are higher than UGoDIT.
> > >
> > > **However, the key message is not that UGoDIT will be better than supervised methods that are well-trained on very large scale matched datasets; instead, the message is UGoDIT-$M$ (with $M\in \\{4,5,6\\}$ sub-sampled images) could achieve competitive PSNRs when compared to an 8000-trained-VarNet (or an 8000-trained-recurrent VarNet) with supervision indicating the favorable trade-offs for a data-scarce methods.**
> > >
> > >
> > > ## Forward operators
> > >
> > > Our MRI forward operators use coil-wise sensitivity maps followed by Fourier transform and undersampling. The forward operators were fixed across methods being compared. As stated in the rebuttal, we used **variable density Cartesian random sampling masks commonly used in compressed sensing MRI**. In addition to the above results, we also included results with uniform Cartesian undersampling (with center of k-space fully sampled) for UGoDIT-6 and recent diffusion-based SITCOM where SITCOM (a high-data method) marginally outperforms UGoDIT.
> > >
> > >
> > > ## Results of ZS-SSL and SSDU in our response to Reviewer eMg2 vs. the ones reported in Table 1 of the ZS-SSL paper.
> > >
> > > For ZS-SSL and SSDU (measurement splitting methods), the hyper-parameters, split ratio, and the network architecture are based on the default settings (from the code base) for the knee fastMRI dataset.
> > >
> > > The results we reported in the rebuttal table were based on running the code base of these methods. We used AF of 4x with the Gaussian 1D sampling mask like the one used in our results of Table 1. Our results for ZS-SSL are consistent with what was reported in Tables 1 and 2 of Self-Guided DIP [33] (IEEE TCI 2025).
> > >
> > > We acknowledge that the PSNRs for ZS-SSL and SSDU are lower than those reported in Table 1 of the ZS-SSL paper as a different dataset (CORP instead of fastMRI) and sampling mask (Uniform 1D instead of Gaussian 1D).
> > >
> > > To fully address the reviewer's concern, we run UGoDIT-6 vs. ZS-SSL **using a uniform 1D sampling mask instead of the Gaussian 1D used in our earlier response**.
> > >
> > >
> > > |Method (MRI task with uniform 1D)|Test-time PSNR|
> > > |-|-|
> > > | UGoDIT-$6$                |   35.88   |
> > > | ZS-SSL (measurement splitting)  |  34.80   |
> > >
> > >
> > > As observed, our method still outperforms ZS-SSL. We note that due to time limit of the rebuttal, we ran the training-data-free version of ZS-SSL, not the ZS-SSL-TL which uses training data. We will include the latter in the revised paper.

---

> > > > ### Comment · Reviewer_oaHC · 2025-08-06
> > > >
> > > > Thanks to the authors for their response. I am still unclear about the response to my first comment. As a reviewer, when I read the paper, what I see is that UGoDIT outperforms all methods, including supervised methods, on all tasks, except for SITCOM-MRI on MRI data, and even that is by a small margin. This statement also includes the new methods included in the rebuttal. Thus, I am not seeing that UGoDIT is merely competitive, which the authors state is their key message, but I am seeing that it is SOTA.
> > > >
> > > > Thus, I would like an explanation as to how a method trained on ~6 randomly selected degraded datasets, can consistently outperform supervised methods on large databases, especially since the latter is not being tested in OOD scenarios. The authors are suggesting this is due to smaller training database size for the supervised methods compared to their original publications, but a ~3dB jump would be surprising if the only change was moving from 8000 training images to ~20000 images. Thus, I would be grateful if the authors can clearly state their underlying hypothesis for their SOTA performance, surpassing supervised methods that are also being tested on matched training conditions.
> > > >
> > > > Finally, as far as I know, E2E-VarNet reports testing and training on all knee MRI datasets, which includes two different types of contrasts/SNRs. The images shown in the current paper only include the high-SNR knee dataset. Can you please clarify if all training/testing for all methods were done on both types of knee data?

---

> > > > > ### Author Response · Authors · 2025-08-07
> > > > >
> > > > > We thank the reviewer for their feedback and are grateful for the consistent engagement. **We first refer the reviewer to our latest response to Reviewer's eMg2 response as we included more ZS-SSL results to fully address the earlier concern about results mismatch.** Next, we attempt to address the latest comments.
> > > > >
> > > > > ## Number of training images in supervised methods
> > > > >
> > > > > In general, we believe that increasing the number of training images could lead to improving the generalization and hence the test performance for supervised reconstruction methods.
> > > > >
> > > > > We verified that our VarNet results are consistent with recent literature. In Table 1 of our paper, the 8000-trained-VarNet obtains 33.06 dB PSNR with Gaussian 1D sampling on 20 test scans which were a subset of the test scans (randomly chosen) used in DDS [12] (ICLR24).
> > > > >
> > > > > In DDS, VarNet was trained with ~25k images (Appendix G.1 in [12]), where they reported test PSNR of 34.15 dB for Gaussian 1D sampling (variable density sampling) as given in Table 1 of [12]. This indicates that using a larger training set improves PSNR for VarNet.
> > > > >
> > > > > The PSNRs for DDS in our work are also similar to those reported in [12]. Indeed, the proposed UGoDIT is competitive against the reported baselines. We hope this clarification helps better explain the results.
> > > > >
> > > > > ## UGoDIT's underlying hypothesis and how it surpasses supervised methods
> > > > >
> > > > > **Our underlying hypothesis is as follows:** UGoDIT inherits the strong implicit bias of DIP-based methods toward natural image statistics while mitigating DIP’s weaknesses (i.e., noise overfitting and slow convergence) by:
> > > > > - (i) Autoencoding regularization and input updates, which enforce encoder-decoder consistency and delay overfitting.
> > > > > - (ii) A multi-decoder architecture during training, where each decoder specializes in reconstructing its corresponding image, while the shared encoder is enforced to extract robust and generalizable features. This over-parameterization results in a stronger encoder that transfers effectively to unseen images.
> > > > > - (iii) An image-specific test-time optimization using the pre-trained encoder and a randomly initialized decoder. The fixed generalizable encoder also alleviates overfitting at test-time.
> > > > >
> > > > > We also remark that:
> > > > > - Recent studies on DIP variants (e.g., aSeqDIP [24] (NeurIPS24), DRP [26] (CVPR23), ES-DIP [34] (TMLR23), Deep Decoder [B] (ICML 2019)) have shown that with careful initialization, constraints, network re-parameterization, and/or regularization, DIP-based methods could also achieve competitive performance as compared to data-intensive methods across multiple tasks. UGoDIT extends this line of work by introducing mechanisms that further enhance robustness and convergence in the low-data unsupervised regime.
> > > > >
> > > > > - There are non-DIP deep learning low-data (or dataless) methods that have achieved competitive results such as those based on measurement splitting with self-supervision (e.g., ZS-SSL [A] (ICLR22)), or implicit neural representation (e.g., STRAINER [37] (NeurIPS24) or DIINN [C] (WACV23)). Although the intuition behind these methods is different than UGoDIT (as they are not DIP), we included the comparisons as these baselines consider the same (or very similar) setting as ours.
> > > > >
> > > > > [A] Zero-shot self-supervised learning for MRI reconstruction, ICLR 2022.
> > > > >
> > > > > [B] Deep Decoder: Concise Image Representations from Untrained Non-convolutional Networks, ICML 2019
> > > > >
> > > > > [C] Single Image Super-Resolution via a Dual Interactive Implicit Neural Network, WACV 2023
> > > > >
> > > > > ## On the training/testing datasets used
> > > > >
> > > > > For our study, the training and testing for UGoDIT and all data-centric baseline methods (including VarNet and Recurrent VarNet) were conducted using both types of knee MRI data available in the fastMRI multi-coil dataset: coronal proton-density (PD) (high SNR/contrast) and coronal proton-density fat-suppressed (PDFS) volumes (lower SNR/contrast). Specifically, our datasets included 100 PD and 99 PDFS volumes, randomly selected as was partially described in the Methods section. This approach ensures that both the training and the testing set has the variability in contrast and SNR present in the fastMRI knee dataset.
> > > > >
> > > > > The images shown in our paper are indeed from the high-SNR PD dataset, but all quantitative results (and model training) used the full set with both contrasts/SNRs. Following the reviewer's questions, we will include other visualizations to highlight this point.
> > > > >
> > > > > ## UGoDIT is SOTA or competitive
> > > > >
> > > > > Our results (in paper and rebuttal) show that UGoDIT mostly achieves SOTA performance under the evaluated settings. **Our aim is to avoid setting unreasonable expectations; however, in the low-data regime, our method clearly pushes the boundaries of the SOTA, which is particularly impactful for data-limited applications.**
> > > > >
> > > > > We hope that our responses address the reviewer's concerns.

---

> > > > > > ### Comment · Reviewer_oaHC · 2025-08-07
> > > > > >
> > > > > > I thank the authors for their rebuttal. I will increase my score accordingly.

---

### Official Review · Reviewer_usw1 · 2025-07-02

**Clarity:** 3
**Significance:** 3
**Originality:** 3
**Rating:** 4
**Confidence:** 3

**Summary:**

UGoDIT is a single-image reconstruction method designed for image recovery and enhancement from compressed signals. This approach combines unsupervised learning and adaptive feature extraction techniques, addressing specific challenges in image restoration by designing a training mechanism and incorporating multiple decoders and a universal encoder.

**Questions:**

1. As a single-image reconstruction method, it is essential to compare the required FLOPs and training time.

2. The paper should include comparisons with the latest image reconstruction methods based on Implicit Neural Representations (INR), such as:

Cai Z. Conv-INR: Convolutional Implicit Neural Representation for Multimodal Visual Signals [J]. arXiv preprint arXiv:2406.04249, 2024.

3. The paper should include generalization validation on larger datasets beyond facial or specialized medical regions, such as for natural images in tasks like super-resolution and compressed sensing. It demonstrates both the efficiency and generalization ability of the proposed method across more diverse and realistic scenarios.

**Ethical Concerns:**

["NO or VERY MINOR ethics concerns only"]

**Final Justification:**

All of my concerns have been addressed. The paper explores an direction for unsupervised image reconstruction, but there is still room for improvement. Therefore, I will maintain my original score. However, The proposed method suffers from two significant limitations:

Limit generalization capability: As noted by the reviewers, the introduction of additional reference images improves reconstruction quality in specific scenarios such as super-resolution of human faces, outperforming baselines like UGoDIT. However, this improvement comes at the cost of reduced performance on the primary evaluation task, MRI reconstruction. This suggests that the method may trade off generalization for performance gains on narrowly distributed data.

Reduce effectiveness of the encoder: The fact that the encoder achieves competitive results even when trained on a minimal amount of data raises concerns about its generalization ability. Ideally, the encoder should benefit from increased data volume by learning more robust and expressive features, leading to better reconstruction. However, the proposed design appears to limit this capacity, indicating a potential flaw in its scalability. I hope the AC will take these limitations into consideration when making the final decision.

**Limitations:**

yes

**Paper Formatting Concerns:**

yes

**Quality:**

3

**Strengths And Weaknesses:**

Advantages of UGoDIT:

Single-image reconstruction: UGoDIT performs image reconstruction and enhancement from compressed signals, making it particularly effective in scenarios where data is limited or compressed.

Unsupervised learning: The method leverages unsupervised learning, which allows it to work without relying on large, labeled datasets for training, reducing the need for ground-truth data.

Improved generalization: The encoder is encouraged to extract more generalizable and robust features due to the multiple decoders, ensuring better reconstruction performance during both training and testing.

Disadvantages of UGoDIT:

Out-of-distribution (OOD) performance variability: UGoDIT's performance on OOD test cases varies across tasks. Specifically, it shows lower OOD performance for tasks like MRI compared to methods like aSeqDIP, limiting its application for certain medical imaging tasks.

Task-dependent performance: While UGoDIT performs well on certain tasks like super-resolution (SR), it does not consistently outperform other methods like aSeqDIP on all tasks, such as MRI, indicating that its strengths may vary based on the specific image recovery task.

The experimental results of the paper show that when the number of images exceeds six, there is no significant improvement in model performance. This indicates that the enhancement from this training approach is limited and lacks strong generalization capability.

---

> ### Author Rebuttal · Authors · 2025-07-31
>
> We'd like to first thank the reviewer for their feedback. We are glad that the reviewer acknowledges the limited-data setting of our work and its fully unsupervised nature.
>
> ## (D1) OOD performance varies across tasks: Lower OOD performance for MRI compared to methods like aSeqDIP.
>
> We thank the reviewer for this comment. We agree that, as shown in Appendix A, UGoDIT performs well on some OOD tasks (e.g., super-resolution) but shows reduced OOD performance for MRI.
>
> We believe this variability stems from two factors:
>
> 1. **Encoder specialization:** UGoDIT learns a shared encoder that captures features from the training degradations. When the test distribution deviates substantially (e.g., training on knees, testing on brains), the learned encoder may not fully generalize.
>
> 2. **Role of autoencoding regularization:** The autoencoding term stabilizes training but also biases the encoder toward the statistics of the training images. This can limit adaptability under severe distribution shifts, whereas per-image methods like aSeqDIP can adjust their weights to the new domain.
>
> While this is a limitation, we see it as an avenue for future work — for example, exploring encoder fine-tuning at test time or hybrid strategies that combine UGoDIT’s transferable features with adaptive refinement. We will discuss this point explicitly in the revised manuscript.
>
> ## (D2) While UGoDIT performs well on certain tasks like super-resolution (SR), it does not consistently outperform other methods like aSeqDIP on all tasks.
>
> We agree with the reviewer that UGoDIT’s performance varies across tasks. However, compared to aSeqDIP and other dataless methods, Figure 3 shows that UGoDIT achieves **significant improvements in convergence speed** while also providing higher reconstruction quality: approximately **+3 dB** PSNR for MRI, **+1 dB** for SR, and **+2 dB or more** for NDB.
>
> To support this claim, we report the number of optimization iterations needed to achieve 30dB across the three task, comparing UGoDIT-6 and aSeqDIP. We note that, at inference, the number of parameters to optimize in UGoDIT (with $M=6$) is lower than aSeqDIP as the encoder in our case is fixed.
>
> | Method                                     | iterations to achieve 30dB (MRI)  | iterations to achieve 30dB (SR)  | iterations to achieve 30dB (NDB) |
> |--------------------------------------------|---------------------|-------------------------------|-|
> | UGoDIT                                   | 10   | 120                  | 650 |
> | aSeqDIP |   450  | 325       | 1750  |
>
> When compared to data-centric baselines such as SITCOM, UGoDIT returns slightly lower PSNR but does so **without requiring any pre-trained model** or access to large datasets of clean images, making it more applicable in low-data settings.
>
>
> ## (D3) The experimental results of the paper show that when the number of images exceeds six, there is no significant improvement in model performance. This indicates that the enhancement from this training approach is limited.
>
> We agree that our experiments show that performance tends to saturate when the number of training images exceeds six. This suggests that UGoDIT is particularly well-suited for the **very low-data regime**, where only a handful of degraded images are available — a setting where many existing methods struggle.
>
> We hypothesize that the saturation effect arises because the encoder quickly captures the shared structure present in the training images, and adding more examples beyond this point yields diminishing returns. This is consistent with our goal of designing a method that is **effective even under severe data scarcity**.
>
> This property also makes UGoDIT appealing for domains where only a few measurements are accessible (e.g., specialized scientific imaging tasks such as black hole observations). We plan to explore such applications in future work.
>
>
> ## (Q1) As a single-image reconstruction method, it is essential to compare the required FLOPs and training time.
>
> We thank the reviewer for this comment. At test-time, UGoDIT is indeed a single-image reconstruction method.
>
> We report the training time and FLOPs at training for the SR task, and compare it with aSeqDIP. All methods were run for 4000 iterations.
>
> | Method            | time (sec) | GFLOPs|
> |-------------------|---------------------|--------|
> | UGoDIT-4 (training)        | 384  | .0143        |
> | UGoDIT-5 (training)        | 415  | .0152        |
> | UGoDIT-6 (training)        | 472  | .0174        |
> | UGoDIT-4 (testing)         | 62  |  .0062        |
> | UGoDIT-5 (testing)         | 64  |  .0064        |
> | UGoDIT-6 (testing)         | 66  |  .0066        |
> | aSeqDIP                    | 142  | .01004       |
>
>
> As observed, the more training images, the higher run-time and FLOPS for UGoDIT as we have more decoders to train when $M$ increases. We are at most less than four times the run-time when compared to aSeqDIP. As for testing, we only require less than half the run-time as aSeqDIP since the optimization in UGoDIT takes place w.r.t. only the decoder, whereas aSeqDIP optimizes the entire network.
>
> Moreover, our run-time and FLOPs are significantly lower than training diffusion models or supervised methods. For example, as per Figure 1 of [A], diffusion models require more than 17 GFLOPs.
>
> [A] Improving Training Efficiency of Diffusion Models via Multi-Stage Framework and Tailored Multi-Decoder Architecture, CVPR 2024
>
> ## (Q2) Comparison and discussion with latest INR methods such as [B].
>
> We thank the reviewer for pointing us to this baseline. **We note that the INR-based method we compare with (i.e., STRAINER [37]) is from NeurIPS 2024**.
>
> Additionally, in this rebuttal, we include comparisons with three more baselines (see our response to Reviewer eMg2 (W4+W5)).
>
> We could not find a code base for [B], and due to the limited time of the rebuttal, we couldn't code it from scratch. However, we promise the reviewer to include discussion (and/or results) of [B] in our revised paper.
>
> [B] Conv-INR: Convolutional Implicit Neural Representation for Multimodal Visual Signals, ArXiv 2024
>
> ## (Q3) Generalization validation on larger datasets.
>
> We thank the reviewer for this suggestion.
>
> For more diverse OOD evaluation, we report the results of an FFHQ-trained-UGoDIT-6 using **100 ImageNet test images** and **20 CBSD65 test images** for the SR task.
>
> | Method            | Test-time Avg. PSNR on 100 ImageNet images |
> |-------------------|------|
> | FFHQ-trained-UGoDIT-6     | 25.67  |
> | ImageNet-trained-UGoDIT-6          | 25.89  |
>
> | Method            | Test-time Avg. PSNR on 20 CBSD65 images |
> |-------------------|------|
> | FFHQ-trained-UGoDIT-6     | 26.98  |
> | CBSD65-trained-UGoDIT-6          | 27.13 |
>
>
> We would like to clarify that we include an evaluation in **Appendix A.2**, where we test an FFHQ-trained-UGoDIT-6 on **20 natural test images from ImageNet** for the super-resolution task in Figure 7 (with visual examples in Figure 10).
>
> The results demonstrate that UGoDIT maintains strong performance across different datasets. We will include these in the revised paper.

---

> ### Comment · Reviewer_usw1 · 2025-08-05
> **To authors**
>
> Thank you for your response. I still believe that the lack of discussion and comparison with relevant INR-based approaches is a key limitation of this work. Such as
>
> [1] Single Image Super-Resolution via a Dual Interactive Implicit Neural Network (WACV 2023)
>
> [2] Implicit Neural Networks with Fourier-Feature Inputs for Free-breathing Cardiac MRI Reconstruction (2023)
>
> are directly related and should been considered for comparison or discussion.
>
> Second, if the authors attribute the saturation effect to the encoder quickly capturing shared structures in the training images, as stated — “We hypothesize that the saturation effect arises because the encoder quickly captures the shared structure present in the training images, and adding more examples beyond this point yields diminishing returns.” — then a more meaningful experimental design would be to train the encoder on a broader and more diverse set of natural images beyond just faces and mri, with increasing dataset sizes, rather than only evaluating on natural images.

---

> > ### Author Response · Authors · 2025-08-06
> >
> > We'd like to thank the reviewer for their response. Below we attempt to address the remaining concerns.
> >
> > ## Discussion and comparison with INR-based approaches: [1] Single Image Super-Resolution via a Dual Interactive Implicit Neural Network (WACV 2023) and [2] Implicit Neural Networks with Fourier-Feature Inputs for Free-breathing Cardiac MRI Reconstruction (2023)
> >
> > We thank the reviewer for sharing these baselines.
> >
> > **Regarding [1]**: Please see the results below from the Urban100 dataset used in Table 1 of [1] (which is an INR prior-free method termed DIINN) on the super-resolution task.
> >
> > | Method|Test-time PSNR (SR with $\times4$) on Urban100|
> > |-|-|
> > |ImageNet-trained-UGoDIT-$6$ |25.21 |
> > |DIINN [1] |24.49 (from Table 1 in [1])|
> >
> >  As our method is DIP-based, we chose aSeqDIP (NeurIPS 2024) and Self-Guided DIP (IEEE TCI 2025) to be our main dataless comparison baselines as these method showed competitive performance on multiple tasks. For INR, in our original submission, we chose STRAINER (NeurIPS 2024) which as it is a multi-head prior-based INR-based method that operates in a similar setting to our method in terms of using very small amount of training data.
> >
> > **Regarding [2]**: Thank you for sharing baseline [2], which is an INR-based method for dynamic MRI. We'd like to clarify that while UGoDIT may have the potential to be extended to dynamic imaging, [2] operates in frame by frame and there is not clear way to consider it for static MRI. We will include this discussion in the "limitations and future work" section.
> >
> > ## Second, if the authors attribute the saturation effect to the encoder quickly capturing shared structures in the training images, then train the encoder on a broader and more diverse set of natural images beyond faces, with increasing dataset sizes, rather than merely evaluating on natural images.
> >
> > We thank the reviewer for the comment and suggestion. We note that, in our rebuttal, we train once with six images from ImageNet and once with six images from CBS65 (both not faces dataset and are broader in general), and evaluate with 100 and 20 test images (also not faces), respectively. We will consider evaluating with increasing dataset sizes in the revised paper.
> >
> > Just to further clarify, when the reviewer says "merely evaluating on natural images", does the reviewer mean that we train with natural images and evaluate on non-natural images (such as MRI data)? Or visa versa? If yes, this will be considered a form of an OOD experiment where the training and testing distributions are extremely different.

---

> > > ### Comment · Reviewer_usw1 · 2025-08-07
> > > **More comments**
> > >
> > > Thank you for your response. I have read your rebuttal, and it aligns with my current understanding.

---

> > > > ### Author Response · Authors · 2025-08-08
> > > >
> > > > We thank the reviewer for their response and would be happy to address any other questions or concerns during the remaining time in the discussion period.

---

### Official Review · Reviewer_eMg2 · 2025-07-02

**Clarity:** 3
**Significance:** 3
**Originality:** 3
**Rating:** 5
**Confidence:** 5

**Summary:**

This work proposes UGoDIT: a DIP-based method that uses a small number of corrupted measurements for training. The method uses  M corrupted images at train time to learn one shared-encoder + M independent decoders, with the additional autoencoder constraint (this is different from DIP). At test time, a new sample is optimized by solving for a new deoder (and keeping the original encoder as fixed).

The method is compared to other variants of DIP, as well as to end-to-end and generative methods trained on large sets of clean images across compressed sensing MRI, super-resolution, and non-linear deblurring. Results indicate that UGoDIT can match or outperform performance of all other approaches.

Strengths and Weaknesses
Strengths:
The paper is easy to follow and the idea is presented in a conceptually simple way. The work contextualizes the different options depending on the amount/quality of available training data, and compares to a number of these combinations. Results are presented over several different imaging inverse problems tasks, including out-of-distribution experiments. Several ablation experiments are performed and discussed.

**Questions:**

Please refer to the Strenghts and Weaknesses section for a list of questions and concerns. Specifically, my main questions are regarding training on clean vs. corrupt, comparison to measurement splitting, and more training details.

**Ethical Concerns:**

["NO or VERY MINOR ethics concerns only"]

**Final Justification:**

I have updated my score from borderline reject to borderline accept. I have also increased my confidence from 4 to 5. I have left detailed comments in response to the authors.

Update Aug 6: I have increased my score to 4 to 5 and increased significance from fair to good.

**Limitations:**

Please refer to the Strenghts and Weaknesses section for additional limitations that should be discussed. Specifically, training memory scales with number of reference images at test time.

**Quality:**

3

**Strengths And Weaknesses:**

Strengths:
The paper is easy to follow and the idea is presented in a conceptually simple way. The work contextualizes the different options depending on the amount/quality of available training data, and compares to a number of these combinations. Results are presented over several different imaging inverse problems tasks, including out-of-distribution experiments. Several ablation experiments are performed and discussed.

Weaknesses:
- It seems like the main contribution of the work is to use a shared-encoder with M decoders at test time. The auto-encoding self-consistency was previously proposed by aSeqDIPin Ref 24.

- The justification on page 5 for the self-consistency is not really clear. Unless I am misunderstanding, the minimum of Eq 2. is z_i = A^T y_i, i.e. do nothing. It is not really clear how this formulation avoids the need for early stopping, which is the original issue with DIP.

- The key idea of the method seems to be unrelated to the fact that the training is over corrupt images. For example, and as another form of ablation, what would happen if M clean images were used to train the shared encoder? Wouldn't this represent the best-case learning? In addition, what is the distinction at test time of the new sample? What would happen if it were included as the (M+1)'th sample at training time instead? Would performance improve at the cost of compute?

- There is no mention or comparison to any of the works that use measurement splitting as a form of self-supervision, even though some of these have been directly applied to DIP. Like my comment above, it seems to me like measurement splitting could be used here as well.

[1] Noise2Self: Blind Denoising by Self-Supervision  Joshua Batson, Loic Royer

[2] Self-supervised learning of physics-guided reconstruction neural networks without fully sampled reference data
Burhaneddin Yaman, Seyed Amir Hossein Hosseini, Steen Moeller, Jutta Ellermann, Kâmil Uğurbil, Mehmet Akçakaya

[3] Zero-Shot Self-Supervised Learning for MRI Reconstruction  Burhaneddin Yaman, Seyed Amir Hossein Hosseini, Mehmet Akçakaya


- There is no discussion regarding test-time training methods, such as the following reference:
[4] Test-Time Training Can Close the Natural Distribution Shift Performance Gap in Deep Learning Based Compressed Sensing Mohammad Zalbagi Darestani, Jiayu Liu, Reinhard Heckel Proceedings of the 39th International Conference on Machine Learning, PMLR 162:4754-4776, 2022.

- It seems like the training requirements scale with the number of images used. This is not discussed as a limitation.

- It is unclear why the method performs better than methods trained on order of magnitude clean images. If this is indeed the case, then this should be emphasized and discussed.


- no details are shared on how the M images were picked. this seems important given the message of appendix  A

- Appendix F figure 13 does not seem to indicate that increasing M helps. While the mean increases, the variance does not always shrink, there are outliers, and none of the results would likely be significant. Furthermore, the difference is less than 0.5 dB. The plot also starts at M=4 (what does it look like for M=1,2,3 ?)

Appendix I: Very few details are given regarding the training setup. The paper does not describe how the train/test sets were formed (are only 20 images used for validation?) The code does not indicate what should be run to replicate the different results, and does not appear to include any of the MRI experiments. Regarding the MRI experiments, how were the slices chosen for training and testing? Appendix I makes it sound like different slices from the same volume were used in training and in testing? Is that the case?

- Almost no images are shown except in the appendix (and all MRI images in the paper are upside-down). The few that are shown are very small; Are the ROIs of Fig 16 the same central image just copied onto the top-left? It would be more helpful to make the ROI larger and increase the scale of the error image.

---

> ### Author Rebuttal · Authors · 2025-07-31
>
> We are glad that the reviewer finds our paper ''easy to follow'' and ''presented in a conceptually simple way'', and acknowledges the several imaging tasks, OOD experiments, and the several ablation studies.
>
> ## (W1) Main contribution is the shared-encoder with $M$ decoders. Self-consistency was proposed by aSeqDIP.
>
> Indeed, our main contribution is the shared-encoder architecture jointly trained with $M$ decoders on degraded images. This design learns transferable encoder weights that significantly accelerate and improve DIP-based reconstruction at test time.
>
> While we extend the “auto-encoding self-consistency” formulation in aSeqDIP [24] to our multi-decoder setting, UGoDIT differs in three key ways:
> - aSeqDIP operates on per-image and learns no transferable prior; UGoDIT learns a shared encoder across $M$ degraded inputs.
> - UGoDIT’s encoder is fixed at test time, enabling faster reconstructions and better generalization (Figures 2-3, and Appendix B). See also our response to Reviewer usw1 (D2).
> - The multi-decoder structure enables task-specific learning (where each task is an inverse problem), accelerating convergence at test time.
>
> Thus, we think that extending self-consistency to a transferable, low-data regime with shared encoder weights is non-trivial and impactful, especially when clean data is unavailable, and it has led to significant improvements in PSNR and convergence.
>
>
> ## (W2) Clarity on Eq. (2) minimizer and early stopping.
>
> This is a two-part concern, and we clarify both points below:
>
> **Regarding Eq. (2):** The objective in Eq. (2) is jointly minimized over encoder $\phi$, the decoders $\psi_i$, and the inputs $z_i$. Eq. (2) is the same as Eq. (1) but with the autoencoding hard constraint over the training set that serves to mitigate overfitting. Since CNNs (like a U-Net) are known to denoise or autoencode images, they serve as an explicit constraint to restrict the solution space. The reviewer’s suggestion that $z_i = A^T y_i$ is the minimizer assumes a fixed network or identity mapping. In contrast, $g_{\psi_i} \circ h_\phi$ is a deep network trained to balance measurement consistency and autoencoding regularization. The input $z_i$ is updated based on the network’s current parameters, and its convergence is determined by the evolving encoder–decoder mapping.
>
> **Regarding early stopping:** The need for early stopping in DIP arises from the risk of overfitting to noise or undesired points in the null space of $A$ when optimizing over all network weights per image. In UGoDIT, the autoencoding term $\|\|g_{\psi_i} \circ h_\phi(z_i) - z_i\|\|_2^2$ helps enforce input–output consistency, acting as a regularizer that stabilizes PSNR curves over longer iterations. As shown in **Appendix C** (Figure 11), this reduces the need for early stopping.
>
> ## (W3) [MAIN] (1) Training with $M$ clean images; (2) Including test image as the (M+1)'th sample at training time.
>
> We appreciate the reviewer's important suggestions. These will help improve our work.
>
> UGoDIT is designed specifically for the **low-data regime without clean images**. However, we agree that training the shared encoder on clean images is a valid comparison and may represent an upper bound as we show next.
>
> If clean images are available, we first need to modify the training objective since there would no longer be forward operators $A_i$ or inverse problems during training. The optimization reduces to a pure multi-decoder autoencoder, where Eq.(3a) becomes:
>
> $\phi, \psi_i \leftarrow \arg \min_{\phi,\psi_i} [\sum_{i\in[M]} \|\|g_{\psi_i}\circ h_\phi(z_i) - z_i\|\|^2_2 ],$ ... (\*)
>
> where $z_i = x_i$ are the clean ground truth images.
>
> To test this idea, we compare UGoDIT trained on degraded images vs. a multi-decoder autoencoder trained on clean versions of UGoDIT's degraded images for the SR task.
>
> | Method                                     | Training  | Test-time PSNR  |
> |--------------------------------------------|---------------------|-------------------------------|
> | UGoDIT                                   | 6 degraded images   | 30.75 $\pm$ 0.58                  |
> | Multi-decoder AE (clean training) Eq. (\*) | 6 clean images      | 31.02 $\pm$ 0.67       |
>
> As observed, test-time PSNR for the auto-encoder with clean data is *slightly* better that UGoDIT. This means that training on $M$ clean images could also serve as a prior for test-time reconstructions. The decoder for clean training is initialized at random and then optimized to reconstruct the test image.
>
> Although this represents a different setup from UGoDIT, we will include this ablation study in the revised paper.
>
> As for the second part of the question, we hypothesize that including the test image as the ($M+1$)-th sample during training will not improve performance compared to our approach of learning a prior and testing on the image separately. This is supported by the results of Figure 1, where the average training curve is lower than the testing curve. The reason is that during training, the network parameters are optimized to solve multiple inverse problems, while at test time the decoder parameters are optimized (with the help of the frozen pre-trained encoder) to solve a single image-specific problem. Furthermore, including the test image as the ($M+1$)-th changes the problem from generalizing to unseen measurements to directly fitting all available data, which—as the reviewer correctly notes—comes at a higher computational cost. UGoDIT, in contrast, is designed for inference on unseen data using a fixed encoder. That said, we believe that the reviewer’s suggestion is insightful, for which we leave a deeper investigation to future work.
>
>
> ## (W4+W5) [MAIN] Comparison with measurement splitting and test-time training.
>
> We thank the reviewer for pointing us to these unsupervised learning baselines. Please see below comparison for the task of MRI.
>
> |Method (MRI task)|Test-time PSNR|
> |-|-|
> | UGoDIT                |   36.07 $\pm$ 0.34   |
> | SSDU [A]  (measurement splitting)                             |  33.25 $\pm$ 0.45   |
> | ZS-SSL [B] (measurement splitting)                                   |  33.47 $\pm$ 0.66   |
> | TTT [C] (test-time with default 300 training images)     | 33.81 $\pm$ 0.45   |
>
> As observed, our method returns the best results as compared to the suggested baselines. We did not have sufficient time to compare to [D], but we promise to include it in a revised version.
>
> [A] (SSDU) Self-supervised learning of physics-guided reconstruction neural networks without fully sampled reference data, MRM 2020
>
> [B] (ZS-SSL) Zero-Shot Self-Supervised Learning for MRI Reconstruction, ICLR 2022
>
> [C] (TTT) Test-Time Training Can Close the Natural Distribution Shift Performance Gap in Deep Learning Based Compressed Sensing, ICML 2022
>
> [D] Noise2Self: Blind Denoising by Self-Supervision, ICML 2019
>
>
> ## (W6) Why does UGoDIT outperform high-data methods?
>
> We thank the reviewer for this question. We believe there are two factors:
>
> **Strong inductive bias of DIP-like architectures:**
> Untrained convolutional networks already capture useful low-frequency image structures (specifically when overfitting is mitigated), UGoDIT leverages this property while learning a transferable encoder that further enhances these representations. For example, see the visualizations in Figure 5 of aSeqDIP [24].
>
> **Regularization through shared encoder and multi-decoder training:** Our training encourages the encoder to learn features that generalize across tasks, while the multi-decoder setup reduces gradient interference and overfitting, resulting in representations that are robust even without large datasets.
>
> We note that we do not claim that UGoDIT will always outperform high-data methods. In settings where large-scale clean datasets are available and well-matched to the test domain, supervised or generative models can achieve higher performance (such as SITCOM vs. our method in Table 2).
>
> Our key message is that UGoDIT offers **competitive performance in the low and degraded data regime**, where such large clean datasets are not available.
>
> ## (W7) [MAIN] Selection of the $M$ training images.
>
> We'd like to thank the reviewer for this comment. **The training and testing images were selected at random** including for the MRI experiments. We refer the reviewer to our response to Reviewer 2CSh (W2+Q3) for more details and experiments about the performance of UGoDIT using different training sets over the same 100 test images.
>
> In our response to Reviewer usw1 (Q3), we report the results of an FFHQ-trained-UGoDIT-6 using **100 ImageNet test images** and **20 CBSD65 test images** for the SR task (additional OOD evaluation).
>
> ## (W8) Figure 13 results for $M\in\\{1,2,3\\}$.
>
> We extend the results of Figure 13 to the following table with $M\in \\{1,2,3\\}$. We note that $M=1$ trains a one-decoder autoencoder and then fixes the weights, so the performance might highly depend on the selected training image.
>
>
> |$M$ in UGoDIT (NDB)|PSNR|
> |-|-|
> |1|29.07 $\pm$ 0.52|
> |2|29.40 $\pm$ 0.56|
> |3|29.80 $\pm$ 0.45|
> |4 (from Figure 13)|30.02 $\pm$ 0.40|
>
> As observed, the performance slightly degrades when $M$ becomes smaller. We will include this result in the revised version.
>
> ## (W9) Training/testing code and setup. Evaluation with $>20$ test images.
>
> Thank you for the comment. We added more details in the anonymous readme file and included the MRI code.  See our response to Reviewer 2CSh (W2+Q3) for evaluation over **100 test images**.
>
> In our response to Reviewer usw1 (Q3), we report the results of an FFHQ-trained-UGoDIT-6 using **100 ImageNet test images** and **20 CBSD65 test images** for the SR task (additional OOD evaluation).
>
> ## (W10) Visualizations in the main text. ROIs. MRI orientation.
>
> We will include visual examples in the main text and fix the orientation of the MRI images. We will also enlarge the ROI (central region) crops and the error maps.

---

> > ### Comment · Reviewer_eMg2 · 2025-08-04
> > **Increasing my score slightly**
> >
> > Thank you for such a comprehensive response to my comments. I believe you have addressed almost all my concerns, except for a few. Thus I am increasing my score from borderline reject to borderline accept. These are my remaining concerns:
> >
> > - Regarding the minimum of the loss:
> > "Since CNNs (like a U-Net) are known to denoise or autoencode images, they serve as an explicit constraint to restrict the solution space." --> this is also the case for the original DIP, which will eventually overfit (i.e., will return A^T y). I do not see how this is any different in your setup and it seems misleading in the "intuition" section. Instead, it seems more reasonable to lean on the experimental results which appear to show that overfitting is slower because of the inclusion of multiple corrupted images at training time.
> >
> > "The reviewer’s suggestion that $z_i = A^T y_i$ is the minimizer assumes a fixed network or identity mapping." --> It does not assume a fixed network or identity mapping; rather, it assumes that there exist at least one (probably multiple) solutions where D(E(z)) \approx A^T y. This is plausible since D, E, and z are trained jointly.
> >
> > - Regarding including the (M+1)th image at test time
> > I really don't understand your explanation. Can this not simply be tested? Here is the explicit question. Suppose I train with M images and test on the (M+1)'th image, call the error on the test image E1. Now I train with all M+1 images (therefore there is no distinction between train and test. Call the error on the (M+1)'th image E2. How does E1 compare to E2? This would show the benefit of the fixed encoder at train time (other than compute, which I agree but can also be explicitly be shown).
> >
> > - Regarding comparison to measurement splitting and high-data methods:
> > What network architecture and how many parameters were used for measurement splitting? I have a hard time believing these results and I want to make sure it is a truly fair comparison. The arguments about "inductive bias" also apply to these methods so I have a hard time buying it. And the fact that you initialize z=A^Ty means that it is essentially the same as these other methods that take A^Ty as an input.

---

### Official Review · Reviewer_2CSh · 2025-07-03

**Clarity:** 1
**Significance:** 2
**Originality:** 2
**Rating:** 5
**Confidence:** 4

**Summary:**

The paper introduces UGoDIT, which uses a shared convolutional encoder ( encoding set of layers of a DIP U-Net) and disentangled convolutional decoders (decoding set of layers of a DIP U-Net). During the training phase, UGoDIT reconstructs $M$ measurements, thereby capturing a data-driven prior and learning transferable weights beneficial at test-time. At test time, the pre-trained encoder is kept frozen while optimizing the decoder parameters in order to recover a clean / noise-free image. The approach is evaluated on accelerated MRI reconstruction, super-resolution, and non-linear deblurring tasks and shows superior performance over existing baselines.

**Questions:**

1. Can the authors provide more analysis on the out-of-distribution capabilities of UGoDIT?
2. Can the authors also provide visualizations or insights on the quality of features learned in the shared encoder?
3. How does the set of training images impact reconstruction quality? For example, Figure 15 (supplementary) exhibits a clear bias towards skin tone in training and evaluation sets.
4. Can the authors comment on the stability and impedance to overfitting noise over a long number of iterations of UGoDIT?

**Ethical Concerns:**

["NO or VERY MINOR ethics concerns only"]

**Final Justification:**

The idea presented in the paper is interesting, with theoretical insights from multi-task learning and recent literature from neural fields.  I would argue that the paper provides limited empirical evidence, making it difficult to thoroughly understand how the proposed method performs in a wide range of inverse problems and compares to competitive baseline methods. Increased my score.

**Limitations:**

Yes. However, please include Limitations in the main paper and not in the supplementary.

**Quality:**

2

**Strengths And Weaknesses:**

__Strengths__:

1. The paper addresses overfitting to noise and high computational cost at inference for deep image prior-based methods by learning generalizable features in the shared encoder layers, resulting in better reconstruction quality.
2. The paper provides a thorough and extensive evaluation. It includes comparisons with relevant baselines and provides rigorous ablation studies.  A clear empirical effect of design decisions is presented in the paper.
3. The papers include experiments to highlight the method’s effectiveness and demonstrate competitive, if not better, performance while using subsampled/degraded to baseline methods, which use fully sampled / clean data.

__Weakness__:
1. Lack of theoretical justification: While the paper shows empirical evaluations, it fails to mention any theoretical insights or analysis as to why sharing the initial encoder layers would learn better feature learning. The authors should detail a more rigorous theoretical intuition beyond what's currently mentioned in section 3.2.
2. Bias in training set: From Figure 15 (supplementary), it seems that the training set and test set exhibit a bias towards properties such as skin tone in face images, which may lead the shared encoder to learn favorable features. Further, MRI examples also include similar structure and intensity patterns. The paper does not discuss the performance of UGoDIT when the training set differs substantially from the test set, while comprising a high variance in structure and texture of samples within the same class/task.
3. The paper is overly verbose and lacks brevity. The paper can perhaps tidy up the repeating equations across the main text and further refine the overall process of UGoDIT and its underlying intuition.
4. Small test sets: The paper tests the approach on 20 images in the test set, which does not convey sufficient information to assess the approach's generalizability and superior performance over baselines.

---

> ### Author Rebuttal · Authors · 2025-07-31
>
> We thank the reviewer for their comments and feedback. We are glad that the reviewer finds that our paper addresses overfitting to noise and high computational cost at inference for DIP-based methods, and acknowledges that we provide a thorough and extensive evaluations. As suggested by the reviewer, we will include the the limitations in the main body of the paper.
>
> ## (W1) Theoretical justification to why sharing the initial encoder layers would learn better feature learning beyond what's currently mentioned in section 3.2.
>
> We thank the reviewer for this valuable comment. While our discussion in Section 3.2 provided intuitive motivation, we now offer a justification from the literature of **multi-task representation learning**. In our setting, each degraded image corresponds to an image-specific inverse problem, which can be interpreted as a separate task.
>
> Our setup aligns with the multi-task learning framework studied in [A, B], where a shared representation (encoder $h_\phi$) is learned jointly across multiple tasks, each with its own function (i.e., task-specific decoder $g_{\psi_i}$ for $i \in [M]$). Specifically, Theorem 3 in [A] and Theorem 2 in [B] demonstrate that this structure (i) reduces generalization error across tasks and (ii) promotes robust and transferable feature learning in the shared encoder. Importantly, these theoretical results rely on the structural assumption of a shared encoder with task-specific heads.
>
> In contrast, the single-encoder, single-decoder setup (as in Eq. (5)) violates this assumption, since the decoder receives gradients from multiple tasks simultaneously. This prevents specialization and introduces **gradient interference** — a known challenge in multi-task learning [C]. Consequently, the theoretical benefits shown in [A, B] do not extend to the single-decoder case which motivates our proposed multi-decoder method. We will include this discussion in the revised paper.
>
>
> [A] A Model of Inductive Bias Learning, JAIR 2000
> [B] The Benefit of Multitask Representation Learning, JMLR 2016
> [C] Gradient Surgery for Multi-Task Learning, NeurIPS 2020
>
> ## (W2+W4+Q2) Is there bias in training set? Impact of the selection of training set. Evaluation on more than 20 test images. Performance of UGoDIT when the training set differs substantially from the test set.
>
> **We would like to emphasize that we selected the training and testing images completely at random**. This includes the set of images in Figure 15, our in-distribution (Section 4), and out-of-distribution (Appendix A) evaluations. This means that there is no intentional, skin-tone, or favorable bias.
>
> We note that the selection of images in our OOD experiments is done such that the training and testing sets are substantially different. For example, the faces training images are the degraded versions of the ones in the first row of Figure 15 while the testing images are the ones in Figures 8 (a bird) and 9 (a front view of a cinema building).
>
> To fully address the comment, we evaluate six models of UGoDIT-6, each trained with a distinct set of randomly selected training images $Y_i$, and baselines using the same **100 test images** for the SR task. Distinct training sets mean that for any $i$ and $j$, $Y_i \cap Y_j = \emptyset$.
>
> | Method       | Test-time Avg. PSNR over 100 test images |
> |--------------|------------------|
> | UGoDIT-6 with $Y_1$       | 30.64 $\pm$ 0.24         |
> | UGoDIT-6 with $Y_2$       | 30.57 $\pm$ 0.31         |
> | UGoDIT-6 with $Y_3$       | 30.65 $\pm$ 0.27         |
> | UGoDIT-6 with $Y_4$       | 30.58 $\pm$ 0.25         |
> | UGoDIT-6 with $Y_5$       | 30.74 $\pm$ 0.16         |
> | UGoDIT-6 with $Y_6$       | 30.95 $\pm$ 0.13         |
> | STRAINER-6       | 29.72      $\pm$ 0.65         |
> | DPS       | 25.63       $\pm$ 0.67         |
> | SITCOM       | 30.81       $\pm$ 0.82         |
>
>
> As observed, the averaged PSNR of our method ranges between 30.58 dB to 30.95 dB, i.e., the variability using different small training sets is very small (less than 0.4 dB). This indicates that the in-distribution results of UGoDIT is not highly sensitive to the selection of training images.
>
> In our response to Reviewer usw1 (Q3), we report the results of an FFHQ-trained-UGoDIT-6 using **100 ImageNet test images** and **20 CBSD65 test images** for the SR task (additional OOD evaluation).
>
> For MRI, we refer the reviewer to our additional results using additional sampling patterns in our response to Reviewer oaHC (W4+Q3).
>
> We will include both results to the revised paper.
>
> ## (W3) The paper is verbose; equations are repeated, and the description of UGoDIT and its intuition should be streamlined.
>
> We thank the reviewer for this helpful comment. In the revision, we will refine the writing near the optimization formulation of UGoDIT and the alternating updates of Eq.(3) by consolidating related derivations, and move secondary details to the appendix.
>
> We agree with the reviewer that equations (3) to (5) have similarities, especially in terms of using a measurement consistency term and an autoencoding term. We will update the writing to reduce redundancy. That said, we respectfully clarify that the equations serve distinct purposes: Eq. (3) introduces the alternating updates to solve Eq. (2). Eq. (4) defines the test-time optimization. Eq. (5) introduces the baseline with the shared decoder.
>
> ## (Q1) Can the authors provide more analysis on the OOD capabilities of UGoDIT?
>
> We thank the reviewer for this question. We include a detailed out-of-distribution (OOD) evaluation in **Appendix A**, where we assess UGoDIT’s performance when tested on distributions that differ from the training set. Specifically:
> - In **Appendix A.1**, we train UGoDIT on knee MRI scans and test it on brain scans. We observe that performance degrades compared to in-distribution cases. See Figure 6.
> - In **Appendix A.2**, for the super-resolution task, we train on FFHQ (dataset of faces) and test on ImageNet (see Figures 8 and 9). UGoDIT shows better convergence and improved final PSNR over standalone DIP (aSeqDIP), demonstrating robustness under natural image distribution shifts.
>
> We will clarify and emphasize these findings more clearly in the revised version.
>
>
> ## (Q2) Can the authors provide visualizations or insights on the quality of features learned in the shared encoder?
>
> We thank the reviewer for this question. It is indeed important to shed more light on the learned features.
>
> In **Appendix D**, we include a study analyzing the features of a UGoDIT encoder trained on MRI data. We analyze the encoder outputs at various layers and compare them with those from a single-decoder baseline.
>
> Our analysis shows that the shared encoder in UGoDIT tends to learn smoother and more structured intermediate representations that are more robust to input variations.
>
> As for visualizations, we note that features at the output of the trained encoder are of much lower dimension and do not typically hold semantic meanings. We will add more context to further clarify this point.
>
>
> ## (Q4) Can the authors comment on the stability to overfitting noise over a long number of iterations of UGoDIT?
>
>
> We thank the reviewer for this question. We partially address this in **Appendix C**, where we evaluate UGoDIT’s robustness to noise overfitting across different values of the regularization parameter $\lambda$, using 10000 iterations. Additionally, in **Appendix A.1**, we run UGoDIT (and other variants) for 30000 iterations in the OOD setting.
>
> We would like to clarify that the autoencoding term $\|\|g_{\psi_i} \circ h_\phi(z_i) - z_i\|\|_2^2$ acts as an implicit constraint that encourages the output of the NN (i.e., the reconstruction) to remain close to the input. This promotes stability during optimization and mitigates overfitting to noise in the measurements. For this reason, we hypothesize that even with substantially more iterations, PSNR curves will plateau rather than degrade, as the network converges to a solution that satisfies both the data fidelity and autoencoding terms.
>
> Due to NeurIPS restrictions on including external links, we were not able to share plots for running UGoDIT for more than 30000 iterations. However, we promise the reviewer that we will include experiments with more iterations in the revised version to further empirically support this claim.

---

> > ### Comment · Reviewer_2CSh · 2025-08-04
> >
> > Thanks to the authors for addressing my concerns. I've increased my score from borderline reject to borderline accept as the authors have addressed some of my concerns.
> >
> > I do feel that the paper could provide more rigorous experimentation, analysis of UGoDIT's performance variations given different kinds of training image sets, and visualization of the underlying feature space learned by the shared encoder to convey the intuition and effect of using a shared encoder more clearly. Further, due to limited evaluation in the paper and rebuttal, it's not completely clear as to how UGoDIT performs against competitive baselines reported in the paper.

---

> > > ### Author Response · Authors · 2025-08-06
> > >
> > > We thank the reviewer for engaging in the discussion. We are glad that the reviewer decided to increase their score from borderline reject to borderline accept, acknowledging that we have addressed some of their concerns. Below, we attempt to address the remaining ones.
> > >
> > > ## Provide more experimentation, analysis of UGoDIT's performance variations given different kinds of training image sets.
> > >
> > > We would like thank the reviewer for this comment. In the revised paper, we will include the additional experiments from the rebuttal where we evaluated six UGoDIT models trained on six different training sets.
> > >
> > > We will repeat these experiments for other tasks (MRI and non-linear deblurring) in the revised paper in order to gain more insights about the capabilities and limitations of UGoDIT.
> > >
> > > ## Visualization of the underlying feature space learned by the shared encoder to convey the intuition and effect of using a shared encoder more clearly.
> > >
> > > As the encoder's feature space is typically much smaller than the image dimension, we are afraid that the outputs of the encoder would have no semantic meanings and therefore may not be useful to measure the learned features. **That said, following the reviewer's suggestion, we will look at the output of each layer (as was similarly done in Appendix D) from the pre-trained encoder's stack to see if we can get more insights visually about the learned features.**
> > >
> > > ## Evaluation in the paper and rebuttal against baselines reported in the paper.
> > >
> > > We would like to thank the reviewer for this comment. It would be great (as this would help us improve our paper) if the reviewer could share (or elaborate more) about an evaluation setting that they think would further show the capabilities and limitations of UGoDIT and how it performs against the 13 baselines considered in the original submission and the rebuttal.
> > >
> > > If the reviewer means larger test sets or other datasets, we would like to clarify that in the rebuttal, we included additional results with 100 test images from ImageNet in our response to Reviewer usw1 (Q3).

---

> > > > ### Comment · Reviewer_2CSh · 2025-08-07
> > > >
> > > > I appreciate the authors extensive efforts during the rebuttal. I will incorporate all findings and update my score accordingly.

---

> > > > > ### Author Response · Authors · 2025-08-08
> > > > >
> > > > > We thank the reviewer for their response and would be happy to address any other questions or concerns during the remaining time in the discussion period.

---

### Note · Authors · 2025-08-12

Dear Area Chair,

First and foremost, we would like to thank all reviewers for their time and feedback. We were encouraged by the positive assessments highlighting "the thorough and extensive evaluation" (**Reviewer 2CSh**), "conceptual simplicity of the presentation and the significance of our method" (**Reviewer eMg2**), "the improved generalization" (**Reviewer usw1**), and "the idea of sharing the encoder" (**Reviewer oaHC**). Most reviews were truly helpful in terms of exploring additional ablation studies/settings of our proposed method.

We have provided a thorough rebuttal to address the concerns raised by the reviewers, mainly regarding

- Ablation studies such as the impact of the selection of the training images.
- The number of test scans or images (which we increased from 20 to 100).
- The recommended additional five baselines (ZS-SSL, SSDU, TTT, Recurrent VarNet, and DIINN) and two datasets (CBSD65 and CORPD) where we showed that our method either outperforms others or achieve competitive results under the considered settings.
- Further clarification of our MRI setup.

**As a result, all reviewers showed satisfaction with our rebuttal and either had initial positive score (Reviewer usw1), increased their score (Reviewers eMg2 and 2CSh), or promised to increase it (Reviewer oaHC).**

In the revised paper, we will include the additional baselines in the main body of the paper (Figure 3 and Tables 1 and 2) and the related work discussion.

Most of the other discussions and studies will be incorporated in the Appendix such as
- Training with $M$ clean images (**Reviewer eMg2**),
- the impact of selecting different training sets (**Reviewer 2CSh**),
- reporting FLOPs and training times (**Reviewer usw1**), and
- how much of the encoder arm needs to be trained over the $M$ images (**Reviewer oaHC**).


Thanks,

Authors

---

### Decision · Program_Chairs · 2025-09-17

**Decision:**

Accept (poster)

**Comment:**

In this paper, the authors present a variation of a Deep Image Prior designed to work in the setting where there are noisy measurements available for a small number of sample images.  In this approach, the model contains a shared encoder and multiple image-specific decoders.  During training time, the shared encoder and image-specific decoders are jointly optimized.  During test time, the shared encoder is held fixed and only an image-specific decoder is optimized to fit the measurements.  The method is shown to outperform baselines on a variety of linear and nonlinear image restoration tasks, along with thorough ablation studies.  The proposed approach has an elegance to it and makes significant progress in addressing challenges of generative models (high data requirements upon training) and deep image priors (overfitting to noise and non-specificity for specific signal classes) when applied to inverse problems.  The work is likely to be of interest to the community.